# RATIONAL IRRATIONALITY: EVALUATING LLMS IN GAMES WITH STRATEGIC BEHAVIOR DISCREPANCIES

## ABSTRACT

Large language models (LLMs) are increasingly deployed in complex decision-making environments. Consequently, evaluating their strategic reasoning abilities is becoming increasingly important. A growing body of research investigates their performance in multi-objective settings, often based on or inspired by game theory, with evaluation typically focusing on the models' ability to align with theoretical expectations. This paper shifts the focus to evaluating the alignment between LLM behavior and human strategic thinking by analyzing LLM responses in a well-established game theory testbed. We revisit three notable games—ROCK, PAPER, SCISSORS (RPS), the CENTIPEDE GAME (CG), and the TRAVELER'S DILEMMA (TD), all of which are characterized by substantial discrepancies between empirical human behavior and theoretical predictions. For each game, we record the choices made by LLM agents and compare them with historical data from human subject experiments to uncover commonalities and particularities in their underlying strategic reasoning patterns. Our results indicate that LLMs are, in general, more aligned with game-theoretical expectations and show limited sensitivity to game hyperparameters. In RPS, most LLMs imitate rational behavior, but perform sub-optimally. In CG, likewise, LLMs adopt rational strategies, learning from past interactions. Finally, in TD they cooperate toward a better payoff, adopting, however, a more prudent strategy plan than humans. [1]

## 1 INTRODUCTION

As large language models (LLMs) increasingly operate as autonomous agents in interactive environments, understanding their decision-making behavior is becoming more essential. Thus, LLMs must be evaluated beyond linguistic competence—specifically, they must be evaluated on how they reason, plan, and act. However, traditional evaluation approaches (Wang et al., 2019b;a; Hendrycks et al., 2021; Srivastava et al., 2023; Suzgun et al., 2023) focus on language-related tasks, *e.g.*, language generation, task-specific performance, and question-answering. In response, newer evaluation frameworks have broadened their scope to cover, among other aspects, high-level reasoning (Zheng et al., 2024; Jin et al., 2023), interactive task execution (Chalamalasetti et al., 2023; Smith et al., 2024), cognitive assessment (Gu et al., 2025; Momentè et al., 2025), and safety benchmarking (Zhang et al., 2025; Yao et al., 2024), offering a more comprehensive view of LLMs' agentic capabilities.

A newly emerging area of research in evaluating LLMs is the exploration of their strategic decision-making abilities. This can be conveniently framed within the context of game theory (GT, Von Neumann, 1928; Nash Jr, 1951). GT provides a robust framework for modeling strategic interactions between rational agents, each aiming at maximizing their payoff in a shared environment. Central to GT is the concept of rationality, where agents are assumed to act in a way that maximizes their individual welfare, *i.e.*, their own objective. Additionally, they further adjust their strategy according to the *common knowledge assumption* (Fujiwara-Greve, 2015, §2.3): the agents know that their opponents are rational, they know that their opponents know that, and so on, *ad infinitum*. This theoretical framework provides a solid foundation for analyzing strategic decision-making, yet its assumptions often fail to fully capture real-world behaviors. Indeed, it has been observed that humans might deviate from purely rational decision-making where witnessed behavior often diverges from the equilibrium strategies predicted by classical GT (Güth et al., 1982; Andreoni and Miller, 1993;

---

[1] Our code is available at `https://anonymous.4open.science/r/games-C1AC`.

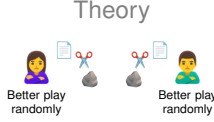

Figure 1: **ROCK, PAPER, SCISSORS**. *Left*: R beats S, S beats P, P beats R, they tie otherwise. *Left-to-center*: The payoffs are $\omega \in (1, +\infty]$ for a win, $1$ for a tie, and $0$ for a loss. *Center-to-right*: The best strategy, according to classic GT, is to randomize the strategies. *Right*: In practice, winners tend to replay their strategies, and losers to change them.

Rosenthal, 1981; Walker and Wooders, 2001; Basu, 2007). These observations prompted economists to explore GT outside of its rational boundaries and to propose more elaborate explanations of human behavior involving aspects not covered by the classical theory, such as altruism or risk-aversion.

Nevertheless, much of the existing work evaluating LLMs in strategic settings focused on their ability to reach equilibrium or maximize payoffs within theoretical frameworks. Compared to this majority, less attention has been paid to the alignment of LLMs with empirical human reasoning (Aher et al., 2023; Ross et al., 2024; Filippas et al., 2024; Akata et al., 2025). Yet, such comparisons are crucial for assessing the extent to which LLM decision-making aligns with or departs from human reasoning.

In this paper, we aim to address this gap by directly comparing LLM strategic behavior with human performance in classic GT experiments. Specifically, we investigate the behavior of LLM agents in three types of games: the evolutionary ROCK, PAPER, SCISSORS, the sequential/extensive-form CENTIPEDE GAME, and the one-shot/normal-form TRAVELER'S DILEMMA. These games model a variety of real-world situations ranging from economics (Edgeworth, 1925; Basu, 2007; Binmore, 2005) to biology (Hopkins and Seymour, 2002; Binmore, 2005), and they are fundamental in that they highlight human (or animal) behavioral patterns that defy the predictions of classical GT. In evolutionary ROCK, PAPER, SCISSORS (RPS, Zhou, 2016), large populations of players poorly randomize their plays, orbiting around the fixed-point equilibrium condition derived in the GT framework. In CENTIPEDE GAME (CG, Rosenthal, 1981; Binmore, 1987) and TRAVELER'S DILEMMA (TD, Basu, 1994), GT rationality predicts that players should converge toward a highly Pareto inefficient equilibrium, which, in practice, is rarely chosen. These deviations from the classical GT predictions reveal the limitations of the model in capturing the complexity of real-world decision-making. By comparing LLM performance in these games to documented human behavior, we aim to assess the degree to which LLMs exhibit rational strategies in line with GT, or whether their decision-making is more aligned with human-like reasoning patterns. This comparison allows us to provide insights into the nature of LLM rationality in complex, multi-objective scenarios.

**Design principles.** We execute our investigation by mirroring canonical human-subject protocols and matching the original rules and payoffs to ensure comparability. In this way, our evaluation of LLMs remains valid relative to established experimental baselines. Additionally, we account for the sensitivity of LLMs to changes in problem formulation. Unlike human participants, models often exhibit heightened variability when prompts are altered, even slightly. To capture this, we introduce multiple verbalizations for the action labels of the games. We then report results aggregated across verbalizations, providing an explicit measure of prompt-induced variance.

**Empirical overview.** Two general trends emerge from our investigation. First, LLMs show, in prevalence, a preference for strategic play leaning toward GT rationality, and less risky planning. Second, they display limited adaptability to payoff-related hyperparameters, which play little to no role in steering LLM reasoning, even in special game environments such as the one investigated in this work. Looking at the single games, the picture is more heterogeneous. In RPS, most models imitate rational strategies, and less frequently, human patterns, failing, in both cases, to consistently maximize their payoff. In CG, LLMs display more rational traits overall, even learning from past interactions. This results in models performing better than the NE strategy, but worse than humans. In TD, most models exhibit a consistent bias toward cooperative play, in misalignment with GT behavior, and only partially aligned with human mixed response. Signs of rationality emerge when examining their strategy, which is more prudent than humans, even in less risky settings.

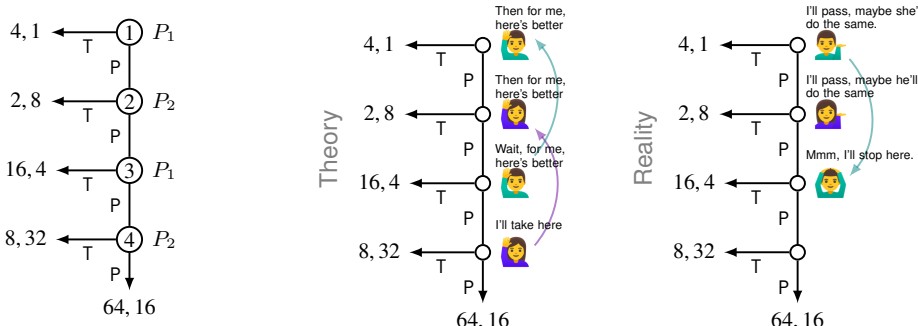

Figure 2: **Centipede Game**. *Left*: At each of the 4 rounds, $P_i$ chooses whether to take (T) or pass (P); the arrows point to the payoffs $\pi_1, \pi_2$ at game termination. *Center*: Backward induction predicts that $P_1$ takes at turn 1, which is the NE. *Right*: In reality, risking and passing is often preferred.

## 2 GAME-THEORETIC RATIONALITY *vs.* HUMAN DECISION-MAKING

Game theory centers on the concept of a **game**, a formalization for a multi-agent interactive environment where rational agents optimize their **payoff** by taking one or multiple actions, defining **strategies**. In this section, we present the three games used in our experiments. For each game, we define a relevant notion of an **equilibrium**, a state in which no agent can unilaterally achieve a higher payoff by changing their strategy. We also review empirical findings from human subject studies, which consistently show that such equilibria are often not reached in practice. For an overview of game types and equilibrium definitions, *cf.* App. A.

### 2.1 ROCK, PAPER, SCISSORS

ROCK, PAPER, SCISSORS (RPS) (Zhou, 2016) is a thoroughly studied game. It consists of two players choosing among three possible strategies, Rock (R), Paper (P), and Scissors (S), and determining their payoffs based on the resulting combination. The payoff functions are equivalent for both players and are illustrated in Fig. 1, using a generic winning payoff parameter $\omega > 1$. In words, a player gets $\omega$ points when winning, 1 point in a tie, and 0 points when losing. Nash Jr (1950) proved the existence of a unique mixed (*i.e.*, probabilistic) strategy Nash equilibrium (NE), $(\frac{1}{3}, \frac{1}{3}, \frac{1}{3})$, consisting in choosing the strategy without any particular preference. When playing multiple games, the opponent cannot therefore exploit any regularity, and is left also to play randomly, therefore reaching an equilibrium.

The RPS game has also been analyzed within the framework of Evolutionary Game Theory (EGT, Weibull, 1997), where a more detailed picture emerges. In this paradigm, a population of $M$ players is repeatedly and randomly paired to play RPS over many iterations. At each step, the distribution of players across the three strategies defines a mixed strategy for the population. Over time, these mixed strategies give rise to a dynamic where players gradually self-organize into cyclic patterns around the game's NE (Weibull, 1997). Analytically, it can be shown that the dynamic converges to the NE when $\omega > 2$, diverges when $\omega < 2$, and stabilizes at the equilibrium when $\omega = 2$.

Wang et al. (2014) tested this prediction empirically by conducting a series of human-subject experiments with varying $\omega$ values. They observed a persistent cyclic behavior in groups playing the evolutionary RPS game. Further, the players appeared to follow a *conditional response* mechanism: repeating the same strategy after a win and switching strategies after a loss. This discrepancy with the classic NE model can be attributed to several factors, including a tendency to imitate opponents' behaviors and the inherent difficulty humans face in generating truly random sequences. While, in theory, such effects should diminish as $\omega$ grows, no significant changes were observed in practice.

### 2.2 CENTIPEDE GAME

CENTIPEDE GAME (CG) is a sequential two-player game introduced in (Rosenthal, 1981). In its most common version, players face two piles of money, each with amounts $m_1$, $M_1$ in $\mathbb{R} \geq 0$, $m_1 < M_1$. On round $t$, player $i$ chooses either to take (T) the larger pile—giving the smaller pile to player

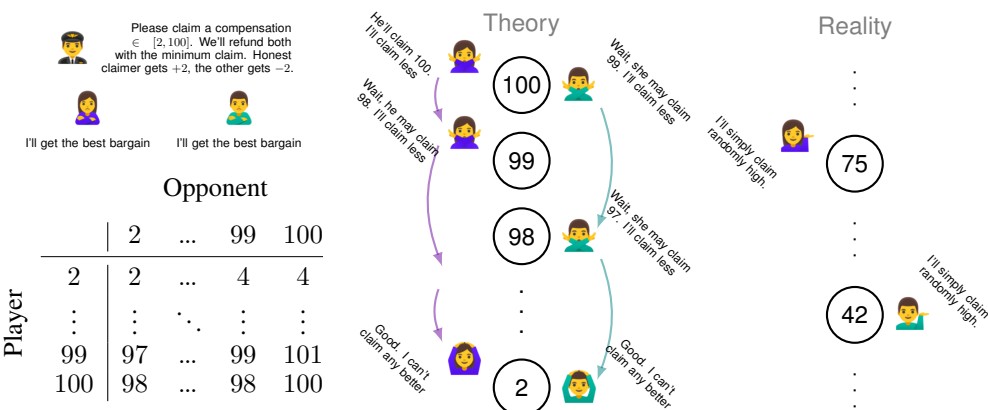

Figure 3: **TRAVELER'S DILEMMA**. *Top left*: Two tourists independently submit compensation claims. *Bottom left*: The lower claim receives a bonus; the higher incurs a penalty. *Center*: GT predicts both players will converge to the lowest possible claim. *Right*: In reality, players tend to make higher claims, favoring better expected payoffs despite potential penalties.

$j$—or to pass (P) the turn to $j$. With each pass, both piles grow by a multiplicative factor, subject to $m_{t+1} < M_t$ to prevent exploitation through passing. The game ends after a maximum number of rounds, known to the players, after which the smaller pile is assigned to the final player to pass receives the smaller pile and the other player the larger. Fig. 2 illustrates a 4-round CG example.

The game is characterized by several possible equilibria, all of which require the first player to immediately take and thus end the game at round 1. The rationality of this behavior can be proven through backward induction. At the final decision node, player $j$ rationally chooses to take the larger pile. It is thus better, for player $i$, to take at the previous round and secure a better payoff. This reasoning continues backward through the game, ending with the first player taking at its first turn. McKelvey and Palfrey (1992) conducted human subject experiments, finding that human players often deviate from this equilibrium condition. In several plays, they effectively passed the turn, and effectively reaching later rounds. The authors motivated this phenomenon assuming a mix of altruistic behaviors and a willingness to take the risk of passing, in hopes of achieving a higher payoff later on.

## 2.3 TRAVELER'S DILEMMA

The TRAVELER'S DILEMMA (TD) was first introduced by Basu (1994) to illustrate a conflict between intuitive reasoning and game-theoretic predictions. The game's context involves two travelers returning from an exotic island, each having purchased identical antiques. Upon arrival, they discover both items have been damaged, and the airline offers to compensate them. Unable to assess the true value, the airline asks each traveler to independently claim a refund amount $n_i \in \{n_{min}, ..., n_{max}\} \subset \mathbb{N}_{\geq 2}$, $i \in \{1, 2\}$ for their own antique. The company then sets the compensation equal to the lower of the two claims, $n_{low}$ (assuming it to be an "honest" amount). Additionally, the honest claimer receives a reward $r > 0$, totaling $n_{low} + r$, while the other receives a penalty of $-r$, reducing their payoff to $n_j - r$. If both claims are equal, no reward or penalty applies ($r = 0$). Given that each traveler aims to maximize their payoff, what would be a rational claim?

The game has symmetric payoffs and is defined as shown in Fig. 3. GT predicts that the NE is at $n_{min}$. Indeed, if both players claim $n_{max}$, player $i$ can improve their payoff by claiming $n_{max} - 1$, earning $n_{max} - 1 + r$. Anticipating this, player $j$ can then claim $n_{max} - 2$ to avoid the penalty and increase their payoff. This backward reasoning continues until both players end up claiming the minimum amount (akin to CG, note, however, that TD is one-shot). Moreover, it can be shown that this NE is unique, even when mixed strategies are permitted (Basu et al., 2011). In the original setup, $n_{min} = 2$, $n_{max} = 100$, and $r = 2$ to highlight the misalignment with human intuition. Indeed, common sense suggests claiming a "large enough" amount, accepting the risk of a small penalty to secure a higher reward. In contrast, claiming the minimum 2, as GT would predict, appears counterintuitive to most people. Capra et al. (1999) conducted repeated rounds of TD with human participants, varying the

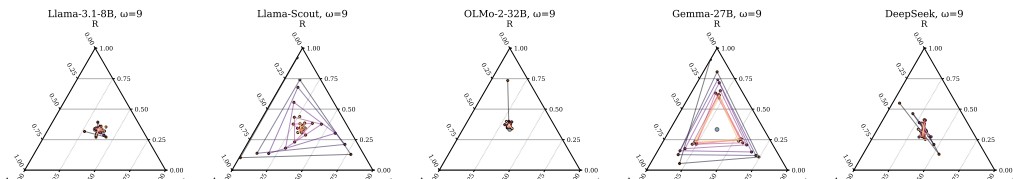

Figure 4: **ROCK, PAPER, SCISSORS– Cyclic Trends.** Each point in the ternary plots represents a distribution over the strategies (R, P, S) played in one round of the RPS game. Points from consecutive rounds are connected by line segments; color encodes time (violet → orange). The central point is the NE (1/3, 1/3, 1/3). Visually, `Llama-4-Scout` and `Gemma-3-27B` exhibit clearer cyclic trajectories around the equilibrium, whereas the other models remain more concentrated near the Nash point.

reward/penalty parameter $r$. Their findings indicate sensibility to high $r$ values, that increase the penalty incurring when the claim is too high, thereby discouraging intuitive, optimistic behavior and pushing players closer to the NE.

## 3 EXPERIMENTS

In this section, we quantify the degree of behavioral alignment between LLMs, humans and GT, in the three games introduced in §2, where theoretical predictions systematically diverge from human play. We evaluate models spanning 3B–671B parameters: `Llama-3.2-3B`, `Llama-3.1-8B`, `Llama-4-Scout`, `OLMo-2-13B`, `OLMo-2-32B`, `Gemma-3-12B`, `Gemma-3-27B`, and `DeepSeek-V3`. Following standard practice in game theory experiments, we assume players may access all information relevant to the games. This means that human subjects may already know such games, and LLMs may have encountered them during pretraining.

We prompt the LLMs with an abstract game description, keeping the payoff structure identical but removing unnecessary information (titles, thematic settings, etc.). To further mitigate verbalization bias, in RPS and CG, we run 10 independent sessions, each with a fresh verbalization of the strategy labels (*i.e.*, using random subsets of Latin letters) to control for semantics embedded in the output symbols. We therefore report results aggregated across verbalizations. When the game comprises multiple rounds, we augment the prompt after each round with the observed history of both players' strategies and realized payoffs. Model descriptions and prompt templates are described in App. D.1 and App. F.

### 3.1 ROCK, PAPER, SCISSORS

We imitate the experimental setup of Wang et al. (2014), and replace human players with LLMs to evaluate their strategic behavior, reducing the number of rounds to 30 and the number of participants to 100 due to computational constraints. We vary the winning payoff $\omega$ over $\{1.1, 2, 4, 9, 100\}$ and for each run all verbalizations as defined above.

In Fig. 4, we present examples of trajectories drawn by LLM populations (*cf.* Fig. 6 for all plots). The trends are similar across all $\omega$. Most models show a fast convergence to the NE, with the exception of `Llama-4-Scout` and `Gemma-27B`, where the frequencies of R, P, and S fluctuate around the NE.

To assess quantitatively the presence of cycles, in Tab. 4, we compute the *cyclic frequency* (Wang et al., 2014; Xu et al., 2013) $f_T = \frac{1}{2\pi T} \sum_{t=0}^{T} \theta(t)$, where $\theta(t)$ is the angle drawn by the trajectory after playing game $t$ and $T$ is the total number of played games. We also test for the hypotheses $f_T \lessgtr 0$. We observe that in the aforementioned models almost all $f_T$ are significant, hinting at the presence of a persistent cyclic behavior. We also register a significant $f_T$ for `Llama-3.1-8B` and `OLMo-32B`, and attribute it to a very narrow cyclic behavior around the NE[2]. Compared to human behavior, the amplitude of the cycle is, in general, larger in LLMs. The Spearman coefficient across all $\omega$ suggests that the winning payoff does not influence the behavior of LLMs.

To assess whether LLMs follow a human-like conditional response strategy (Wang et al., 2014), in Tab. 1, we calculate the probabilities of repeating the same strategy after a win ($W_0$), a tie ($T_0$), or

---

[2]The $f_T$ metric operates with angles, and does not take into account the proximity to the NE.

Table 1: **ROCK, PAPER, SCISSORS– Conditional Response.** Probability of replaying the previous round's strategy. Most LLMs, similar to humans (Figure 2, Wang et al., 2014), are likely to repeat their strategy when they have won in the previous round ($W_0$), compared to when they have lost or tied ($L_0$ and $T_0$). However, $W_0 < 1/3$, confirming that they prefer to change strategy, overall. For the extended table reporting standard deviations across different verbalizations, see Tab. 3.

| $p_0\%$ | $\omega = 1.1$ | | | $\omega = 2$ | | | $\omega = 4$ | | | $\omega = 9$ | | | $\omega = 100$ | | |
|---|---|---|---|---|---|---|---|---|---|---|---|---|---|---|---|
| | $W_0$ | $T_0$ | $L_0$ | $W_0$ | $T_0$ | $L_0$ | $W_0$ | $T_0$ | $L_0$ | $W_0$ | $T_0$ | $L_0$ | $W_0$ | $T_0$ | $L_0$ |
| Humans | 56.0 | **61.0** | 32.0 | **53.0** | 44.0 | 37.0 | **51.0** | 39.0 | 35.0 | **45.0** | 35.0 | 39.0 | **46.0** | 36.0 | 34.0 |
| Llama-3.2-3B | **1.3** | 0.5 | **1.3** | 1.0 | 0.2 | **1.3** | 1.1 | 0.1 | **1.2** | **1.5** | 0.3 | 1.3 | **1.4** | 0.2 | **1.4** |
| Llama-3.1-8B | **14.9** | 9.5 | 11.2 | **15.2** | 9.1 | 11.0 | **15.7** | 8.7 | 10.9 | **16.1** | 9.8 | 11.4 | **15.0** | 9.3 | 11.1 |
| Llama-Scout | **4.9** | 1.8 | 0.1 | **3.1** | 0.5 | 0.1 | **4.2** | 0.4 | 0.1 | **3.2** | 0.4 | 0.1 | **2.6** | 0.4 | 0.1 |
| Gemma-12B | 34.1 | **71.6** | 37.9 | 31.2 | **77.1** | 35.8 | 30.1 | **79.2** | 32.6 | 35.3 | **80.5** | 38.6 | 48.7 | **77.4** | 52.1 |
| Gemma-27B | **55.9** | 13.1 | 17.3 | **58.8** | 11.6 | 24.2 | **58.7** | 15.1 | 25.0 | **55.4** | 9.1 | 24.1 | **58.2** | 9.6 | 21.5 |
| OLMo-13B | **1.1** | 0.7 | 0.5 | **1.0** | 0.4 | 0.4 | **1.2** | 0.4 | 0.3 | **1.1** | 0.5 | 0.3 | **1.5** | 0.8 | 0.6 |
| OLMo-32B | **11.2** | 9.7 | 5.6 | **9.0** | 6.0 | 3.3 | **10.1** | 7.7 | 4.1 | **10.4** | 8.4 | 3.3 | **12.5** | 9.4 | 4.6 |
| DeepSeek | **7.6** | 0.5 | 0.2 | **10.5** | 0.4 | 0.2 | **8.4** | 0.1 | 0.1 | **8.1** | 0.3 | 0.2 | **5.5** | 0.1 | 0.1 |

a loss ($L_0$). Conditional response requires $W_0 > W_s$ and $L_0 < L_s$, with $s \in \{+, -\}$ indicating the choice of one of the other two available strategies. Both humans and LLMs exhibit a higher likelihood of repeating a previous strategy following a win. However, for almost all LLMs, $W_0$ and $L_0$ consistently stay below $1/3$, suggesting that models, unlike humans, are predisposed to switch strategies regardless of the game's outcome. In Gemma-27B, $W_0 > 1/2$, confirming it to be the only model exhibiting conditional response.

If the alignment with human behavior is only partial, do LLMs act rationally? In Tab. 5, we report the expected payoffs obtained by the models across the played RPS games. Wang et al. (2014) show that, when $\omega > 2$, playing the NE generally grants a better payoff than other strategies, consistent with the evolutionary theory. However, the authors also find that some conditional response configurations perform better than the NE and report that for all $\omega$ values, human groups indeed adjust their responses accordingly and score better payoffs. In LLMs, instead, this phenomenon is not witnessed, and all models tend to underperform with respect to the NE scores. The worst-performing models either adopt biased or (*i.e.*, Gemma-12B) cyclic strategies (*i.e.*, LLaMA-Scout, Gemma-27B).

*In evolutionary RPS, LLMs poorly align with human behavior. They sometimes show cyclic motion, with some models showing a clear convergence to the NE. Additionally, LLMs partially follow a conditional response strategy, and unlike humans, they tend to change strategy even after winning. Finally, most models do not align to GT expectations and underperform in terms of expected payoffs.*

## 3.2 CENTIPEDE GAME

Following the experimental protocol of McKelvey and Palfrey (1992), we instantiate 100 LLM agents and split them evenly into two pools. Across 10 rounds, each agent is matched with a new opponent from the *other* pool, yielding 500 two-player games per condition. We vary the game horizon $H \in \{4, 6\}$. We experiment with two payoff schedules, $(4, 1)$ and $(16, 4)$, as low vs. high stakes variants (larger magnitudes increase the gains from passing and the losses from being taken). For each combination of the above parameters, we run 10 verbalizations of the prompt. Following McKelvey and Palfrey (1992), we present in Tab. 2 the number of games $G_t$ that reached round $t$, along with the proportion $p_t$ of games ending at that round. Overall, human players exhibit lower T rates compared to models, indicating that LLMs tend to adopt a more rational strategy. Notably, Gemma-27B demonstrates a seemingly "altruistic" behavior more aligned with the Pareto efficient outcome (PE, *i.e.*, last round is reached), while Llama-4-Scout and DeepSeek adhere more closely to the theoretical equilibrium predictions. The absolute counts $G_t$ reveal that a substantial fraction of models opt to take at rounds 2 or 3, suggesting a willingness to risk for higher payoffs. Similar patterns are observed in the high-payoff and 6-round settings (*cf.* Tab. 6), where the LLMs do not display significant changes in response to the different game hyperparameters.

In Tab. 7, we additionally report the cumulative distribution $F_t$ of game-ending rounds, comparing the first five games with the last five for each agent. Human players consistently adapt over time, learning

Table 2: **CENTIPEDE GAME: Implied probabilities.** Ratio $p_t$ of games ending in round $t$ among those that reached $t$. We also report the absolute number of games $G_t$ reaching $t$. For human players, the probability of choosing T gradually increases as the game develops, whereas LLMs tend to take earlier. All entries are reported as *mean $\pm$ standard deviation* across verbalizations. See Tab. 6

| $p_t\%$ $(G_t)$ | $p_1$ | $(G_1)$ | $p_2$ | $(G_2)$ | $p_3$ | $(G_3)$ | $p_4$ | $(G_4)$ |
|---|---|---|---|---|---|---|---|---|
| Humans | 7 | (281) | 38 | 261 | 65 | (161) | 75 | (57) |
| LLaMA-3.2-3B | $30.1_{\pm14.6}$ | (500) | $21.8_{\pm12.0}$ | (350) | $26.1_{\pm15.0}$ | (280) | $29.7_{\pm17.5}$ | (218) |
| LLaMA-3.1-8B | $36.8_{\pm5.3}$ | (500) | $38.5_{\pm4.9}$ | (316) | $62.7_{\pm7.3}$ | (195) | $58.7_{\pm7.7}$ | (74) |
| LLaMA-Scout | $61.6_{\pm7.3}$ | (500) | $97.6_{\pm3.0}$ | (192) | $100.0_{\pm0.0}$ | (4) | - | (0) |
| Gemma-12B | $7.0_{\pm3.7}$ | (500) | $98.4_{\pm3.6}$ | (465) | $78.8_{\pm11.7}$ | (7) | $66.7_{\pm57.7}$ | (2) |
| Gemma-27B | $3.7_{\pm8.4}$ | (500) | $14.4_{\pm28.9}$ | (482) | $3.2_{\pm5.6}$ | (422) | $53.5_{\pm35.8}$ | (413) |
| OLMo-13B | $25.8_{\pm12.6}$ | (500) | $30.3_{\pm18.1}$ | (371) | $55.7_{\pm14.0}$ | (269) | $52.5_{\pm16.1}$ | (130) |
| OLMo-32B | $21.1_{\pm10.8}$ | (500) | $49.8_{\pm6.1}$ | (394) | $78.8_{\pm10.9}$ | (200) | $81.9_{\pm5.1}$ | (42) |
| DeepSeek | $52.1_{\pm4.9}$ | (500) | $79.4_{\pm6.6}$ | (239) | $84.3_{\pm9.7}$ | (50) | $96.8_{\pm5.2}$ | (8) |
| NASH EQ. | 100 | | - | | - | | - | |
| PARETO EFF. | 0 | | 0 | | 0 | | 0/100 | |

to take earlier and ending games before reaching equilibrium. This behavior is also witnessed in LLMs, where all models consistently increase their T rates in the last games at almost every turn.

In Tab. 8, we report the players' expected payoffs. As expected, almost all models secure better scores than the NE strategy, but underperform compared to humans, penalized by the LLMs' tendency to take at earlier turns. Notably, Gemma-27B imitates human behavior, achieving better payoffs, while larger models *i.e.*, LLama-Scout and Deepseek are penalized by their overly rational play.

*In CG, LLMs play more rationally than humans in that they exhibit higher overall T rates. Additionally, like humans, they consistently learn to take earlier through experience. The strategy adopted grants payoffs above those obtained with the NE, but below the human strategy.*

## 3.3 TRAVELER'S DILEMMA

We test two settings: a *symmetric* (classic) case where the bonus/penalty $r$ is identical for both players, and an *asymmetric* case with $r_1 \neq r_2$. Following Capra et al. (1999), each player completes a session of $T=10$ consecutive Traveler's Dilemma games to allow adaptation to feedback. We use $N=256$ players split evenly into two pools; in each game, opponents are randomly re-matched across pools (one from each pool). Players choose integer claims in $[n_{\min}, n_{\max}] = [80, 200]$. We vary the bonus/penalty $r \in \{5, 10, 20, 25, 50, 80\}$.

In Fig. 5, we present claim trends by round for humans and LLMs. Most LLMs adapt to the adversary plays and gradually increase their claims toward the Pareto-efficient outcome (both players choose $n_{\max}$). In contrast, DeepSeek tends to track the Nash equilibrium more closely. We also observe that LLMs are largely insensitive to changes in $r$, exhibiting similar trends across the tested values.

In Tab. 9, we present the average claim values in games 1 and 8–10. Humans tend to decrease their average claim in later games, when $r \geq 20$, due to the high penalty. LLMs' behavior is instead consistent with reaching the PE outcome. The exception is represented by DeepSeek, which acts more rationally by decreasing its claims, replicating the pattern already witnessed in previous games. In comparison with humans, the models gain better payoffs in high $r$ settings and worse when the bonus/penalty is low. The latter phenomenon is explained by human initial claims starting higher than the corresponding LLMs' claims.

In the asymmetric case, we follow (Basu et al., 2011) where human participants were asked to play with two bonus/penalty values $r_1, r_2 \in \{10, 80\}$. In Tab. 10, we report the average claims of player 1 for the four possible combinations. Basu (1994) reaffirm the findings of Capra et al. (1999) and extend them to asymmetric scenarios, showing that human players behave more rationally when their $r$ value is higher, regardless of their opponent's. LLMs do not follow, instead, any consistent trend, with half of the models being unaffected by the asymmetry. Notably, some models react to the opponent's payoff increasing their claim, while LLaMA-Scout plays around the NE, in case one of

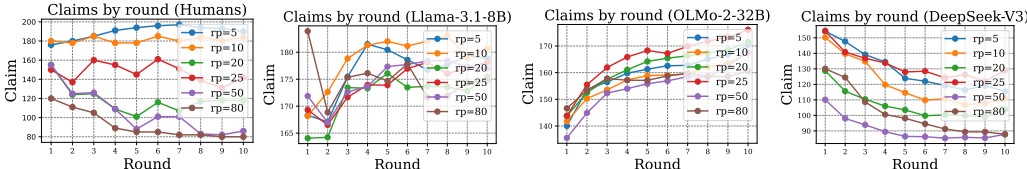

Figure 5: **TRAVELER'S DILEMMA: Claims by round.** Most LLMs adjust their claims based on recent history. Within a session, the average claim often shows a monotonic trend (decreasing or increasing) as the game progresses, similar to human players (leftmost panel, (Capra et al., 1999)). For an extended version, see Fig. 7.

the two $r$ values is high. We attribute the higher variability in the LLM outputs to limited access to game-related information, since the asymmetric case is less studied in the literature.

*In TD, LLMs show model-dependent tendencies, often converging to the PE outcome, unaffected by the reward/penalty values. The lack of flexibility indicates a significant misalignment with human decision-making. This behavior is also reflected in the LLMs' tendency to learn to claim higher.*

## 4 DISCUSSION

The obtained results show that LLMs tend to align more with game-theoretic predictions than with human strategies, though the strength of this alignment depends on the game. In CG, LLMs show superior performance in terms of plain GT payoffs, balancing between a risk-and-pass strategy, and taking at turn 1. The models also seem to grasp the backward induction reasoning behind the game by learning to take earlier. In RPS, the models either try to imitate the NE strategy and, less often, the conditional response. However, they underperform due to the intrinsic complexity of the game posed by the coordination with many players. In TD, there is a general tendency to the PE outcome, and a general sensitivity to the game hyperparameters, probably caused by the large strategic space. For the same reason, the models tend to explore the space toward the PE outcome, but more prudently than humans, especially when the bonus/penalty is low.

Another interesting result is the high standard error of our statistics across verbalizations. Several studies (Aher et al., 2023; Filippas et al., 2024; Phelps and Russell, 2024; Fan et al., 2024) have investigated the effect of prompts on the strategy adopted by LLMs, changing, for example, sensitive information or player intents. In this study, we attempted a more "abstract" approach by removing unnecessary contextual information in the description of the game. We vary, instead, the verbalization of the strategies, which is not guided by a specific design principle but is chosen randomly. The high variability suggests a more thorough study to uncover strategy-dependent playing patterns.

The purpose of our study is to document whether the behavior of LLMs aligns more with that of humans or GT. Due to the special nature of the studied games, *i.e.*, the strong misalignment between theory and common sense, the experiments we conducted can help discriminate between these two poles. There is no particular intent to prioritize one of the two behaviors. Indeed, we might ask ourselves whether the more preferable LLM behavior should be human-like or GT-like. From the viewpoint of GT, this boils down to the social dilemma question: aiming for a globally better but less safe payoff, or a more stable and lower one. Similarly, we do not control for game leakage. The player can be provided with as much information as required – which is even a strict requirement in GT, postulated by the common knowledge assumption – but still choose a different way to play (*cf.*, for instance, the aforementioned ULTIMATUM GAME). If an LLM aligns with GT expectations, then it chooses a more secure, lower payoff, and this is often the case as the experiments show.

## 5 RELATED WORK

Given the extensive research on assessing strategic thinking in LLMs and space constraints, we focus on the studies most pertinent to our work. For an exhaustive overview, we suggest Sun et al. (2025).

Most evaluation efforts have focused on classic two-player games that are well-established in the GT literature. The most prominent example is PRISONER'S DILEMMA (PD, Kuhn, 2024), where two rational players fail to cooperate toward a collectively optimal outcome and "betray" each other, driven by the temptation of a better individual payoff. Several recent studies have explored how LLMs perform in PD scenarios. Some focus on the models' ability to maximize individual rewards (Akata et al., 2025), while others investigate how strategic behavior changes when the model is prompted with contextual information or character-specific details (Lorè and Heydari, 2024), at times revealing altruistic tendencies (Phelps and Russell, 2024; Azaria, 2023) and human-like behavioral biases (Herr et al., 2024). In certain cases, LLMs have exhibited higher levels of cooperation (Brookins and DeBacker, 2023; Fontana et al., 2025) than predicted by standard GT models, suggesting the potential for reasoning beyond traditional payoff-maximization frameworks.

Another studied setting is ULTIMATUM GAME (UG, Harsanyi, 1961) and its variants, in which a proposer decides how to split a fixed sum of money with a recipient. The recipient can either accept the offer—resulting in the proposed split—or reject it, in which case neither player receives anything. In some variants, the recipient has no agency and must accept the offer. The game has long served as a benchmark for studying deviations from GT rationality, as players often demonstrate altruistic or fairness-oriented behavior that departs from strict payoff-maximization. In this regard, UG has been instrumental in probing economic decision-making and social preferences in both humans and artificial agents. Recent studies have examined LLM behavior in this context, testing their responses to abstract formulations of the game (Azaria, 2023; Johnson and Obradovich, 2023; Ross et al., 2024), as well as their adaptability to varying contextual prompts (Phelps and Russell, 2024; Filippas et al., 2024; Fan et al., 2024). In some cases, LLMs have demonstrated sensitivity to superficial details such as the names, genders, or titles of the players (Aher et al., 2023), revealing implicit biases and highlighting the importance of prompt phrasing in shaping strategic behavior.

Beyond these classical games, other studies have revealed additional tendencies in LLMs, including limited coordination capabilities (Akata et al., 2025), a preference for trustful strategies that yield favorable outcomes (Azaria, 2023), and occasional reasoning failures—even in simple or trivial game settings (Lorè and Heydari, 2024). Despite the variability in experimental designs and research objectives, a consistent trend emerges: the most advanced models, particularly those from the GPT (Achiam et al., 2023) and Llama (Touvron et al., 2023) families, exhibit a stronger grasp of game mechanics. These models are capable of interpreting personality traits specified in prompts (*e.g.*, selfishness, altruism) and can adjust their strategic and communicative behavior accordingly (Lorè and Heydari, 2024).

Finally, to support more systematic evaluations, several benchmarks have recently been introduced (Duan et al., 2024; Wang et al., 2024; Shapira et al., 2024; Huang et al., 2025). These benchmarks aim to improve the fairness and consistency of assessments by covering a wider range of game types, from one-shot scenarios to iterative and sequential interactions. Additionally, standardized experimental designs for testing strategic behavior in LLMs have been proposed (Hua et al., 2024), helping to align future work and enable more robust comparisons across studies.

## 6 CONCLUSION

In this work, we have examined the alignment between LLMs and humans in strategic behavior through a game-theoretic perspective. Specifically, we have evaluated LLM performance in three games where empirical findings diverge from theoretical predictions. Our analysis indicates that LLMs tend to imitate rational behavior, blending sometimes with human-like strategies. Moreover, LLMs appear mostly insensitive to numerical payoff-related hyperparameters in prompts, contrasting with typical human behavior.

Our findings provide a preliminary understanding of the alignment between LLMs and human strategic reasoning. Each game studied here deserves a more detailed analysis in future work, as the space of both game- and LLM-related hyperparameters is huge. As a next step, we aim to understand the observed misalignment of LLMs by leveraging insights from game theory literature, considering different behavioral biases (*e.g.*, altruism, risk aversion, etc.), and devising specific phenomenological models to better understand these behaviors.

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

## A    NOTIONS OF GAME THEORY

In this section, we provide a primer in Game Theory useful to understand the main concepts in our work.

**Normal-Form Game (NFG).**    A normal-form game (such as Traveler's Dilemma) models strategic interactions where all players make their moves simultaneously and independently. Formally, an NFG is defined as a triple $\mathcal{G} = (\mathcal{P}, \mathcal{A}, \mathcal{U})$, consisting of a set of players $\mathcal{P}$, their respective strategy sets $\mathcal{A}$, and payoff functions $\mathcal{U}$. The **players**, denoted $\mathcal{P} = \{1, \dots, M\}$, are the agents making decisions in the game. Each player $i$ has an associated **strategy set** $\mathcal{A}_i$, which contains all possible strategies—either discrete or continuous—that the player can choose. All strategy sets are collected in $\mathcal{A} = \{\mathcal{A}_1, \dots, \mathcal{A}_M\}$. Once each player selects a strategy, they receive a payoff determined by their **payoff function** $u_i : \bigtimes_{j=1}^{M} \mathcal{A}_j \to \mathbb{R}$, which depends on the strategy of all

players. Similarly, $\mathcal{U} = \{u_1, \dots, u_M\}$. A combination of strategies $\{a_1, \dots, a_M\}$, where $a_i \in \mathcal{A}_i$, is called a **strategy profile**, representing a complete specification of all players' choices. A strategy profile $a^* = \{a_1^*, \dots, a_M^*\}$ is a **Nash Equilibrium (NE)** if no player can improve their payoff by unilaterally deviating. That is, for every player $i$ and for all alternative strategies $a_i \in \mathcal{A}_i$, it holds that

$$u_i(a^*) \geq u_i(\{a_1^*, \dots, a_i, \dots, a_M^*\}). \tag{1}$$

In such a state, each player's strategy is optimal given the strategies of the other players.

**Extensive-Form Game (EFG).** An extensive-form game (such Centipede Game) can be viewed as a generalization of an NFG, that allows players to take sequential actions across multiple rounds. While a full formalization of EFGs involves several additional concepts beyond the scope of this work, we provide an intuitive yet rigorous overview. The structure of an EFG is commonly represented as a directed tree $\mathcal{K} = (V, E, T)$. Each vertex $v \in V$ corresponds to a decision point assigned to a player, and the outgoing edges $E_v \subseteq E$ represent the available actions at that point. The game begins at the root of the tree and progresses along a path determined by the players' choices, eventually reaching a terminal node $t \in T$. Each terminal node specifies the payoffs for the players based on the actions taken along the path. A player's strategy in an EFG consists of choosing an action at each vertex in the tree where they are required to make a decision[3]. A complete strategy profile is the combination of strategies for all players. As in the normal-form setting, a Nash Equilibrium is a strategy profile where no player has an incentive to unilaterally deviate.

**Evolutionary Game (EG).** An evolutionary game (such Evolutionary Rock, Paper and Scissors) is a rather loose concept encompassing several possible definitions, often applied in GT-based biological evolution dynamics (*i.e.*, Evolutionary Game Theory Weibull (1997)). It models the interaction of a population of **individuals** playing a series of multiple NFGs. At each step of the sequence (*i.e.*, **generation**), the individuals are randomly grouped in pairs and play the same NFG with the same strategy set $\mathcal{A}$ and payoff $u : \mathcal{A} \times \mathcal{A} \to \mathbb{R}$. Based on the received payoff, they can adapt their behavior accordingly for the next generation[4]. The focus in an EG is on the evolution of the entire population summarized in each generation by a **mixed strategy**. Formally, a mixed strategy at generation $g$ is a probability distribution $p^{(g)} = (p_1^{(g)}, \dots, p_{|\mathcal{A}|}^{(g)})$ over the shared strategy set, where each $p_a^{(g)}$ represents the share of individuals in the population playing strategy $a \in \mathcal{A}$. A strategy $a^*$ is said to be **Evolutionary Stable (ES)** if a population in the vicinity of $a^*$ converges to it. Additionally, $\mathcal{U}(a^*, a^*)$ is a NE, therefore $a^*$ is also informally denoted as the NE of the game.

# B   Notions of Language Models

## B.1   Alphabets and Strings

An **alphabet** $\Sigma$ is a non-empty, finite set of elements called **symbols**. A **string** $\boldsymbol{\sigma}$ over alphabet $\Sigma$ is a finite sequence $\boldsymbol{\sigma} = \sigma_1 \cdots \sigma_N$ of symbols, $\sigma_1, \dots, \sigma_N \in \Sigma$. For any alphabet $\Sigma$, let $\Sigma^*$ denote the set of all strings over $\Sigma$. Furthermore, let $|\boldsymbol{\sigma}|$ denote the string's length $N$. Given a string $\boldsymbol{\sigma}$ such that $|\boldsymbol{\sigma}| \geq t$, let $\boldsymbol{\sigma}_{<t}$ denote the string of the first $t-1$ characters of $\boldsymbol{\sigma}$. We write $\boldsymbol{\sigma} \preceq \boldsymbol{\sigma}'$ if $\boldsymbol{\sigma}$ is a prefix of $\boldsymbol{\sigma}'$, and use $\succeq$ and $\succ$ to refer to the relations $\preceq$ and $\prec$ with their respective arguments transposed. The relation $\preceq$ defines a partial order on $\Sigma^*$. Finally, we denote EOS as a distinguished **end-of-string symbol**, which is not a member of the alphabet $\Sigma$.

# C   From Strings to Strategies

While GT players are constrained to choose only the strategies prescribed by the game, an LLM output can span a much wider vocabulary. In the following, we provide a formalization of the relationship between strings generated by LLMs and strategy sets, used in our experiments.

Given an alphabet $\Sigma$, its Kleene closure $\Sigma^*$ is, in short, the set of all possible finite-length strings composed with $\Sigma$. Let, then, $(\Sigma^*, 2^{\Sigma^*}, \mathbb{P}_{\Sigma^*})$ be a **probability space over strings**, with (a countably

---

[3]This is independent of the player actually reaching the vertex in the game.

[4]Other variants are possible, with the individuals playing fixed strategies and adapting the population shares of individuals, instead of their personal behaviors.

infinite) sample space $\Sigma^*$, $\sigma$-algebra $2^{\Sigma^*}$, and a probability measure over the strings $\mathbb{P}_{\Sigma^*}$. Similarly, we define a **probability space over strategies** $(\mathcal{A}, 2^{\mathcal{A}}, \mathbb{P}_{\mathcal{A}})$. Here, $\mathcal{A}$ contains a player's strategies $a_1, ... a_k$ and a symbol $\psi$ denoting an invalid (*i.e.*, not present in the game) strategy[5].

Now, let the **encoding function** $\mathrm{e} : \Sigma^* \to \mathcal{A}$ be a measurable function that maps a string to a strategy. The probability measure $\mathbb{P}_{\mathcal{A}}$ is then the **pushforward measure** induced by $\mathrm{e}$, defined as:

$$\mathbb{P}_{\mathcal{A}}(E) = \mathbb{P}_{\Sigma^*}(\mathrm{e}^{-1}(E)), \quad \forall E \in 2^{\Sigma^*}. \tag{2}$$

Without loss of generality, for $a \in \mathcal{A}$, we can express:

$$\mathbb{P}_{\mathcal{A}}(a) = \mathbb{P}_{\mathcal{A}}(\{a\}) = \sum_{\boldsymbol{w} \in \Sigma^*} \mathbb{1}\{\mathrm{e}(\boldsymbol{w}) = a\} \mathbb{P}_{\Sigma^*}(\boldsymbol{w}) = \sum_{\boldsymbol{w} \in \mathrm{e}^{-1}(\{a\})} \mathbb{P}_{\Sigma^*}(\boldsymbol{w}). \tag{3}$$

where $\mathbb{1}\{\mathrm{e}(\boldsymbol{w}) = a\}$ is an indicator function returning 1 if $\mathrm{e}(\boldsymbol{w}) = a$ and 0 otherwise.

As an example consider the ROCK, PAPER, SCISSORS game. Suppose $\Delta$ is the alphabet used by an LM and $\Delta^*$ is its Kleene closure. Let $\mathcal{A} = \{\mathsf{R}, \mathsf{P}, \mathsf{S}, \psi\}$ and $\mathrm{e} : \Delta^* \to \mathcal{A}$ an encoding function mapping strings to strategies (*e.g.*, through regular expression matching). Then $\mathrm{e}$ induces a partition on $\Delta^*$. In particular, let $S_{\mathsf{R}}, S_{\mathsf{P}}, S_{\mathsf{S}}$ be the sets of strings corresponding to each valid strategy. For example, $S_{\mathsf{R}} = \mathrm{e}^{-1}(\{\mathsf{R}\}) = \{\mathsf{rock}, \mathsf{Rock}, \mathsf{ROCK}\}$.

The measure $\mathbb{P}_{\Sigma}$ assigns probabilities to the strings in $\Delta^*$. Then, the probability of selecting a valid strategy, *e.g.*, $\mathsf{R}$, is given by summing the probabilities of all strings mapped to $\mathsf{R}$ under $\mathrm{e}$:

$$\mathbb{P}_{\mathcal{A}}(\mathsf{R}) = \sum_{\delta \in \Delta^*} \mathbb{1}\{\delta \in S_{\mathsf{R}}\} \mathbb{P}_{\Sigma}(\delta) = \sum_{\delta \in \mathrm{e}^{-1}(\{\mathsf{R}\})} \mathbb{P}_{\Sigma}(\delta) \tag{4}$$

In our work, we use a simple encoding function based on regular expressions to identify strategy choices from model outputs. A string is mapped to one of the strategies $\mathsf{R}$, $\mathsf{P}$, or $\mathsf{S}$ if and only if it exactly matches the regular expression `"I choose (rock|paper|scissors)"`, as specified in the system prompt describing the game. With appropriate prompt design and conditioning (*cf.* App. F for detailed prompt templates), we observe that the probability assigned to the invalid strategy $P_{\mathcal{A}}(\psi)$ —*i.e.*, strings that do not match any valid strategy—is negligibly low. We estimate this leakage using Monte Carlo sampling (2560 samples), and find it to be approximately 0 for RPS, 0.012 for CG, and 0.008 for TD, using `LLama-3.2-3B`. The leakage is even smaller using larger LLMs. Given the small entity of the phenomenon, we handle the rare invalid outputs using a simple heuristic: when an invalid strategy is selected, we resample from the language model output.

### C.1 LANGUAGE MODELS AND PREFIX PROBABILITY

A **language model** $p_{\Sigma}$ is a probability distribution over $\Sigma^*$, where $\Sigma$ denotes an alphabet. It is often useful to represent the probability of a string as a product of single-symbol **conditional prefix probability** distributions, $\overrightarrow{p_{\Gamma}}(\cdot \mid \boldsymbol{\sigma}_{<t})$:

$$p_{\Sigma}(\boldsymbol{\sigma}) = \overrightarrow{p_{\Gamma}}(\mathrm{EOS} \mid \boldsymbol{\sigma}) \prod_{t=1}^{|\boldsymbol{\sigma}|} \overrightarrow{p_{\Gamma}}(\sigma_t \mid \boldsymbol{\sigma}_{<t}) \tag{5}$$

Each conditional prefix probability $\overrightarrow{p_{\Gamma}}(\sigma_t \mid \boldsymbol{\sigma}_{<t})$ is a distribution over the set $\Sigma \cup \{\mathrm{EOS}\}$ and is defined as

$$\overrightarrow{p_{\Gamma}}(\boldsymbol{\sigma}' \mid \boldsymbol{\sigma}) \overset{\text{def}}{=} \mathbb{P}_{Y \sim p_{\Sigma}}[Y \succeq \boldsymbol{\sigma}\boldsymbol{\sigma}' \mid Y \succeq \boldsymbol{\sigma}] = \frac{\overrightarrow{p_{\Gamma}}(\boldsymbol{\sigma}\boldsymbol{\sigma}')}{\overrightarrow{p_{\Gamma}}(\boldsymbol{\sigma})} \tag{6}$$

$$\overrightarrow{p_{\Gamma}}(\mathrm{EOS} \mid \boldsymbol{\sigma}) \overset{\text{def}}{=} \frac{p_{\Sigma}(\boldsymbol{\sigma})}{\overrightarrow{p_{\Gamma}}(\boldsymbol{\sigma})} \tag{7}$$

---

[5]In game theory, each player $i$ can have its own probability space, constructed on its "extended" strategy set $\mathcal{A}_i' = \mathcal{A}_i \cup \{\psi\}$ (*cf.* App. A). In the games under scrutiny in this work, the players are equivalent and have the same set of strategies, thus we omit the extra indices.

The conditional prefix probability is well-defined only when $\overrightarrow{p_\Gamma}(\boldsymbol{\sigma}) > 0$, a standard caveat for conditional probabilities.[6] In turn, the **prefix probability** $\overrightarrow{p_\Gamma}(\boldsymbol{\sigma})$ is the probability that $Y$ has $\boldsymbol{\sigma}$ as a prefix:

$$\overrightarrow{p_\Gamma}(\boldsymbol{\sigma}) \stackrel{\text{def}}{=} \mathbb{P}[Y \sim p_\Sigma] Y \succeq \boldsymbol{\sigma} = \sum_{\boldsymbol{\sigma}' \in \Sigma^*} \mathbb{1}\{\boldsymbol{\sigma}' \succeq \boldsymbol{\sigma}\} \, p_\Sigma(\boldsymbol{\sigma}') \tag{8}$$

This formulation underlies most modern autoregressive language models, where each conditional prefix probability is produced by a learned parametric model.

# D  EXPERIMENTAL SETTINGS

## D.1  PRETRAINED LANGUAGE MODELS

In our experiments, we use pretrained language models from Huggingface, namely `Llama-3.2-3B-Instruct`[7], `Llama-3.1-8B-Instruct`[8], `Llama-4-Scout`[9], `OLMo-2-0325-32B-Instruct`[10], `OLMo-2-1124-13B-Instruct`[11], `Gemma-3-12B-it`[12], `Gemma-3-27B-it`[13], and `DeepSeek-V3-0324`[14]. In our preliminary experiments, we found that varying the sampling temperature ($[0.5, 2.0]$) only marginally changes the strategy distribution of LLMs. Increasing the temperature beyond $2.0$ resulted in significantly more invalid strategies. Thus, we use a default temperature of $1.0$.

## D.2  COMPUTE RESOURCES

All our experiments are conducted with 4 NVIDIA A100 80GB GPUs. For the `DeepSeek-V3` model, we use Together API[15] to obtain the responses. For all the experiments, we estimate a total of roughly one week of compute time.

# E  DETAILED EXPERIMENTAL RESULTS

## E.1  ROCK, PAPER, SCISSORS: ADDITIONAL RESULTS

---

[6]In practice, this holds for softmax-normalized LMs, that assign nonzero probability to all symbols in $\Sigma^*$.

[7]https://huggingface.co/meta-llama/Llama-3.2-3B-Instruct

[8]https://huggingface.co/meta-llama/Llama-3.1-8B-Instruct

[9]https://huggingface.co/meta-llama/Llama-4-Scout-17B-16E-Instruct

[10]https://huggingface.co/allenai/OLMo-2-0325-32B-Instruct

[11]https://huggingface.co/allenai/OLMo-2-1124-13B-Instruct

[12]https://huggingface.co/google/gemma-3-12b-it

[13]https://huggingface.co/google/gemma-3-27b-it

[14]https://huggingface.co/deepseek-ai/DeepSeek-V3-0324

[15]https://www.together.ai/

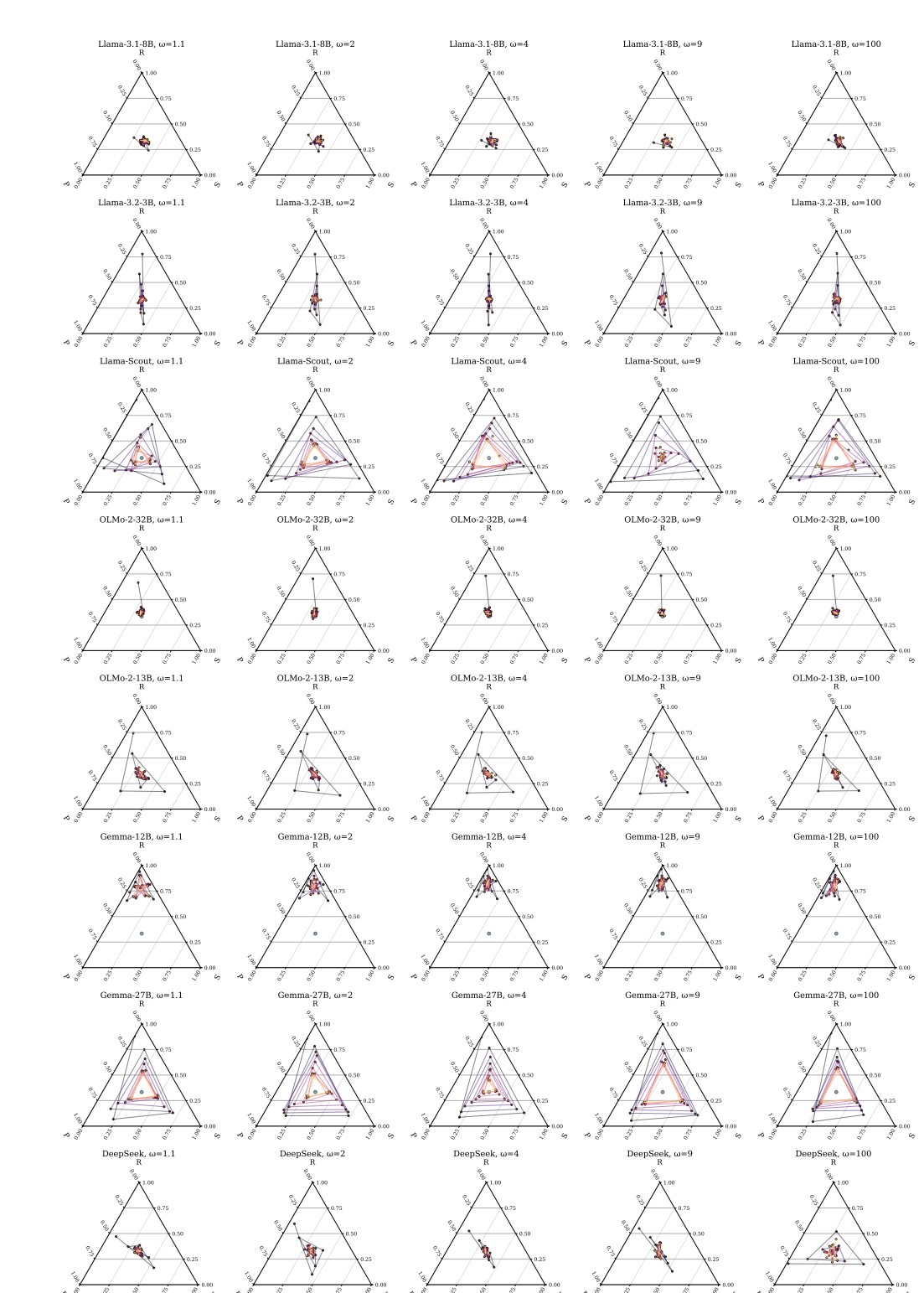

Figure 6: Cyclic trends in RPS across models. Columns (left→right): $\omega \in \{1.1, 2, 4, 9, 100\}$.

Table 3: **ROCK, PAPER, SCISSORS — Conditional response (extended).** Extended version of Table 1, which reports the probability of replaying the previous round's strategy. As in the main table, most LLMs—similar to humans (Wang et al., 2014, Fig. 2)—are more likely to repeat after a win ($W_0$) than after a loss or tie ($L_0$, $T_0$), yet $W_0 < 1/3$ overall. Here we additionally report the *mean ± standard deviation* across different verbalizations for each entry.

(a) $\omega \in \{1.1, 2, 4\}$

| | $\omega = 1.1$ | | | $\omega = 2$ | | | $\omega = 4$ | | |
| --- | --- | --- | --- | --- | --- | --- | --- | --- | --- |
| $p_0\%$ | $W_0$ | $T_0$ | $L_0$ | $W_0$ | $T_0$ | $L_0$ | $W_0$ | $T_0$ | $L_0$ |
| Llama-3.2-3B | $1.3_{\pm0.8}$ | $0.5_{\pm0.3}$ | $\mathbf{1.3}_{\pm0.5}$ | $1.0_{\pm0.7}$ | $0.2_{\pm0.2}$ | $\mathbf{1.3}_{\pm0.7}$ | $1.1_{\pm0.5}$ | $0.1_{\pm0.1}$ | $\mathbf{1.2}_{\pm0.6}$ |
| Llama-3.1-8B | $\mathbf{14.9}_{\pm1.8}$ | $9.5_{\pm1.6}$ | $11.2_{\pm1.4}$ | $\mathbf{15.2}_{\pm2.0}$ | $9.1_{\pm1.4}$ | $11.0_{\pm1.6}$ | $\mathbf{15.7}_{\pm2.8}$ | $8.7_{\pm1.6}$ | $10.9_{\pm1.4}$ |
| Llama-Scout | $\mathbf{4.3}_{\pm2.4}$ | $2.2_{\pm3.0}$ | $0.1_{\pm0.2}$ | $\mathbf{2.4}_{\pm2.3}$ | $0.7_{\pm1.7}$ | $0.1_{\pm0.2}$ | $\mathbf{12.7}_{\pm30.8}$ | $0.5_{\pm1.0}$ | $0.1_{\pm0.1}$ |
| OLMo-13B | $\mathbf{1.1}_{\pm0.6}$ | $0.7_{\pm0.5}$ | $0.5_{\pm0.5}$ | $\mathbf{1.0}_{\pm0.8}$ | $0.4_{\pm0.3}$ | $0.4_{\pm0.4}$ | $\mathbf{1.2}_{\pm0.8}$ | $0.4_{\pm0.3}$ | $0.3_{\pm0.2}$ |
| OLMo-32B | $\mathbf{11.2}_{\pm5.5}$ | $9.6_{\pm6.2}$ | $5.6_{\pm3.7}$ | $\mathbf{9.0}_{\pm3.8}$ | $6.1_{\pm3.3}$ | $3.3_{\pm2.5}$ | $\mathbf{10.1}_{\pm4.4}$ | $7.6_{\pm6.4}$ | $4.1_{\pm2.3}$ |
| Gemma-12B | $42.7_{\pm28.8}$ | $\mathbf{67.7}_{\pm43.6}$ | $35.4_{\pm22.5}$ | $42.2_{\pm28.9}$ | $\mathbf{69.0}_{\pm43.0}$ | $46.3_{\pm28.2}$ | $45.9_{\pm33.6}$ | $\mathbf{69.9}_{\pm43.1}$ | $38.4_{\pm30.7}$ |
| Gemma-27B | $\mathbf{54.9}_{\pm26.4}$ | $17.1_{\pm11.7}$ | $14.3_{\pm10.6}$ | $\mathbf{46.6}_{\pm28.3}$ | $15.8_{\pm12.9}$ | $20.2_{\pm14.6}$ | $\mathbf{53.1}_{\pm24.7}$ | $18.4_{\pm12.8}$ | $23.8_{\pm15.0}$ |
| DeepSeek | $7.7_{\pm4.7}$ | $0.5_{\pm0.5}$ | $0.2_{\pm0.1}$ | $\mathbf{10.6}_{\pm7.1}$ | $0.4_{\pm0.3}$ | $0.2_{\pm0.2}$ | $\mathbf{8.5}_{\pm4.5}$ | $0.1_{\pm0.2}$ | $0.1_{\pm0.1}$ |

(b) $\omega \in \{9, 100\}$

| | $\omega = 9$ | | | $\omega = 100$ | | |
| --- | --- | --- | --- | --- | --- | --- |
| $p_0\%$ | $W_0$ | $T_0$ | $L_0$ | $W_0$ | $T_0$ | $L_0$ |
| Llama-3.2-3B | $\mathbf{1.5}_{\pm0.7}$ | $0.3_{\pm0.2}$ | $1.3_{\pm0.4}$ | $1.4_{\pm0.7}$ | $0.2_{\pm0.2}$ | $\mathbf{1.4}_{\pm0.8}$ |
| Llama-3.1-8B | $\mathbf{16.1}_{\pm1.5}$ | $9.8_{\pm1.2}$ | $11.4_{\pm1.2}$ | $\mathbf{15.0}_{\pm1.3}$ | $9.3_{\pm0.9}$ | $11.1_{\pm1.0}$ |
| Llama-Scout | $\mathbf{3.0}_{\pm1.9}$ | $0.4_{\pm0.9}$ | $0.1_{\pm0.2}$ | $\mathbf{1.8}_{\pm1.8}$ | $0.6_{\pm1.8}$ | $0.1_{\pm0.2}$ |
| OLMo-13B | $\mathbf{1.1}_{\pm0.6}$ | $0.5_{\pm0.4}$ | $0.3_{\pm0.2}$ | $\mathbf{1.5}_{\pm0.9}$ | $0.8_{\pm0.5}$ | $0.6_{\pm0.4}$ |
| OLMo-32B | $\mathbf{10.4}_{\pm4.5}$ | $8.4_{\pm5.1}$ | $3.3_{\pm2.0}$ | $\mathbf{12.6}_{\pm5.7}$ | $9.2_{\pm6.1}$ | $4.7_{\pm2.6}$ |
| Gemma-12B | $51.5_{\pm35.8}$ | $\mathbf{71.9}_{\pm41.5}$ | $53.3_{\pm33.6}$ | $52.1_{\pm28.9}$ | $\mathbf{72.1}_{\pm39.6}$ | $56.6_{\pm30.4}$ |
| Gemma-27B | $\mathbf{38.8}_{\pm28.6}$ | $13.2_{\pm11.0}$ | $17.2_{\pm14.6}$ | $\mathbf{40.8}_{\pm29.7}$ | $14.3_{\pm12.1}$ | $15.9_{\pm14.2}$ |
| DeepSeek | $\mathbf{8.2}_{\pm4.4}$ | $0.3_{\pm0.7}$ | $0.2_{\pm0.2}$ | $\mathbf{5.5}_{\pm1.4}$ | $0.1_{\pm0.1}$ | $0.1_{\pm0.1}$ |

Table 4: **ROCK, PAPER, SCISSORS: Cyclic frequency.** $f_T = \frac{1}{2\pi T}\sum_{t=0}^{T}\theta(t)$, where $\theta(t)$ is the angle drawn by the trajectory after playing game $t$ and $T$ is the total number of played games (*cf.* Wang et al. (2014)). For each model and $\omega$, we report the *mean ± standard deviation* of $f_T$ *across verbalizations*. Spearman's coefficients $\rho$ assess monotonic correlation with $\omega$. Asterisks denote $p < 0.05$ from one-sided *permutation tests* of $H_0 : \mathbb{E}[f_T] = 0$ versus $H_1 : \mathbb{E}[f_T] < 0$.

| $f_T \times 10^{-2}$ | $\omega = 1.1$ | $\omega = 2$ | $\omega = 4$ | $\omega = 9$ | $\omega = 100$ | $\rho$ |
| --- | --- | --- | --- | --- | --- | --- |
| Humans | $3.1$ | $2.7$ | $3.1$ | $2.2$ | $1.8$ | $-0.82$ |
| Llama-3.1-8B | $-5.3_{\pm7.9}$ | $-4.2^*_{\pm5.7}$ | $-4.7^*_{\pm5.9}$ | $-4.9^*_{\pm4.3}$ | $-2.4_{\pm5.5}$ | $0.60$ |
| Llama-3.2-3B | $1.7_{\pm16.4}$ | $0.8_{\pm11.0}$ | $4.7_{\pm9.3}$ | $7.7_{\pm11.0}$ | $4.4_{\pm12.1}$ | $0.60$ |
| Llama-Scout | $-19.7^*_{\pm10.6}$ | $-15.5^*_{\pm13.3}$ | $-15.9^*_{\pm18.4}$ | $-15.9^*_{\pm11.8}$ | $-18.9^*_{\pm13.9}$ | $0.00$ |
| OLMo-13B | $-8.7^*_{\pm8.6}$ | $-6.0_{\pm9.5}$ | $-8.6^*_{\pm8.0}$ | $-7.5^*_{\pm10.2}$ | $-5.0_{\pm8.4}$ | $0.70$ |
| OLMo-32B | $0.7_{\pm4.1}$ | $1.3_{\pm8.3}$ | $-2.0_{\pm7.1}$ | $1.6_{\pm3.6}$ | $-1.9_{\pm5.0}$ | $-0.10$ |
| Gemma-12b | $-3.0_{\pm10.3}$ | $-3.1_{\pm8.5}$ | $-2.4_{\pm7.4}$ | $-1.1_{\pm4.5}$ | $-1.1_{\pm4.3}$ | $0.90^*$ |
| Gemma-27b | $-13.0^*_{\pm13.2}$ | $-14.6^*_{\pm16.2}$ | $-11.8^*_{\pm11.1}$ | $-14.2^*_{\pm17.1}$ | $-10.5_{\pm19.3}$ | $0.50$ |
| DeepSeek | $5.2_{\pm20.5}$ | $-6.1_{\pm20.1}$ | $2.7_{\pm18.7}$ | $1.4_{\pm22.0}$ | $-15.7_{\pm4.8}$ | $-0.70$ |

$^*p$-value $< 0.05$ ($H_1$: mean $< 0$).

Table 5: **ROCK, PAPER, SCISSORS– Average per-round payoff** $\overline{\pi}$. Asterisks (*) indicate values significantly greater than the Nash equilibrium baseline $(a + 1)/3$ under a one-sided sign-flip permutation test ($p < 0.05$; alternative: model > NE). Daggers ($\dagger$) indicate values significantly less than the Nash equilibrium ($p < 0.05$; alternative: model < NE). Wang et al. (2014) do not report exact human payoffs, but state they weakly exceed the NE strategy.

| $\overline{\pi}$ | $\omega = 1.1$ | $\omega = 2$ | $\omega = 4$ | $\omega = 9$ | $\omega = 100$ |
|---|---|---|---|---|---|
| Humans | $> 0.7$ | $> 1$ | $> 1.\overline{6}$ | $> 3.\overline{3}$ | $> 33.\overline{6}$ |
| DeepSeek | 0.713* | 0.986 | 1.625$^\dagger$ | 3.181$^\dagger$ | 31.966$^\dagger$ |
| Llama-3.2-3B | 0.724* | 1.002 | 1.625$^\dagger$ | 3.182$^\dagger$ | 31.887$^\dagger$ |
| Llama-3.1-8B | 0.697 | 1.005 | 1.662 | 3.356 | 32.848$^\dagger$ |
| Llama-Scout | 0.777* | 0.995 | 1.351$^\dagger$ | 2.824$^\dagger$ | 19.581$^\dagger$ |
| OLMo-13B | 0.724* | 0.996 | 1.665 | 3.194$^\dagger$ | 31.705$^\dagger$ |
| OLMo-32B | 0.710* | 1.006 | 1.662 | 3.290 | 32.320$^\dagger$ |
| Gemma-12B | 0.915* | 1.002 | 1.199$^\dagger$ | 1.727$^\dagger$ | 11.958$^\dagger$ |
| Gemma-27B | 0.794* | 1.004 | 1.490$^\dagger$ | 2.356$^\dagger$ | 21.820$^\dagger$ |
| NASH EQ. | 0.7 | 1 | $1.\overline{6}$ | $3.\overline{3}$ | $33.\overline{6}$ |

\* $p < 0.05$    ($H_1$: $\overline{\pi} >$ NE)
$\dagger$ $p < 0.05$    ($H_1$: $\overline{\pi} <$ NE)

## E.2 CENTIPEDE GAME: ADDITIONAL RESULTS

Table 6: **CENTIPEDE GAME– Implied probabilities.** $p_t = e_t / \sum_{j=t}^{n} e_j$. For the $(4, 1)$ payoff schedule we have $m_1 = 1$, $M_1 = 4$; when labeled *H.payoff* we use the high-stakes schedule $(16, 4)$ with $m_1 = 4$, $M_1 = 16$. Cells report *mean $\pm$ standard deviation* across verbalizations; the value $(G_t)$ is the mean number of sequences that reached round $t$. This table is an extended version of Table 2.

| | $p_t\%$ $(G_t)$ | $p_1$ | $(G_1)$ | $p_2$ | $(G_2)$ | $p_3$ | $(G_3)$ | $p_4$ | $(G_4)$ | $p_5$ | $(G_5)$ | $p_6$ | $(G_6)$ |
|---|---|---|---|---|---|---|---|---|---|---|---|---|---|
| **4 Rounds** | Humans | 7 | (281) | 38 | (261) | 65 | (161) | 75 | (57) | | | | |
| | LLaMA-3.2-3B | $30.1_{\pm14.6}$ | (500) | $21.8_{\pm12.0}$ | (350) | $26.1_{\pm15.0}$ | (280) | $29.7_{\pm17.5}$ | (218) | | | | |
| | LLaMA-3.1-8B | $36.8_{\pm5.3}$ | (500) | $38.5_{\pm4.9}$ | (316) | $62.7_{\pm7.3}$ | (195) | $58.7_{\pm7.7}$ | (74) | | | | |
| | LLaMA-Scout | $61.6_{\pm7.3}$ | (500) | $97.6_{\pm3.0}$ | (192) | $100.0_{\pm0.0}$ | (4) | - | (0) | | | | |
| | OLMo-13B | $25.8_{\pm12.6}$ | (500) | $30.3_{\pm18.1}$ | (371) | $55.7_{\pm14.0}$ | (269) | $52.5_{\pm16.1}$ | (130) | | | | |
| | OLMo-2-32B | $21.1_{\pm10.8}$ | (500) | $49.8_{\pm6.1}$ | (394) | $78.8_{\pm10.9}$ | (200) | $81.9_{\pm5.1}$ | (42) | | | | |
| | Gemma-12B | $7.0_{\pm3.7}$ | (500) | $98.4_{\pm3.6}$ | (465) | $78.8_{\pm11.7}$ | (7) | $66.7_{\pm57.7}$ | (2) | | | | |
| | Gemma-27B | $3.7_{\pm8.4}$ | (500) | $14.4_{\pm28.9}$ | (482) | $3.2_{\pm5.6}$ | (422) | $53.5_{\pm35.8}$ | (413) | | | | |
| | DeepSeek | $52.1_{\pm4.9}$ | (500) | $79.4_{\pm6.6}$ | (239) | $84.3_{\pm9.7}$ | (50) | $96.8_{\pm5.2}$ | (8) | | | | |
| **4 Rounds (H. Payoff)** | Humans | 15 | (100) | 44 | (85) | 67 | (48) | 79 | (16) | | | | |
| | LLaMA-3.2-3B | $24.6_{\pm10.9}$ | (500) | $22.2_{\pm12.1}$ | (377) | $26.2_{\pm12.6}$ | (298) | $28.0_{\pm16.5}$ | (228) | | | | |
| | LLaMA-3.1-8B | $36.8_{\pm5.0}$ | (500) | $36.7_{\pm3.9}$ | (316) | $59.1_{\pm7.1}$ | (201) | $56.2_{\pm6.4}$ | (83) | | | | |
| | LLaMA-Scout | $67.2_{\pm6.7}$ | (500) | $96.9_{\pm4.2}$ | (164) | $96.2_{\pm9.9}$ | (5) | $100.0_{\pm0.0}$ | (0) | | | | |
| | OLMo-13B | $26.5_{\pm14.1}$ | (500) | $23.4_{\pm16.3}$ | (368) | $46.0_{\pm22.0}$ | (292) | $41.4_{\pm20.9}$ | (176) | | | | |
| | OLMo-2-32B | $22.5_{\pm10.7}$ | (500) | $50.6_{\pm8.0}$ | (388) | $75.0_{\pm10.2}$ | (194) | $80.5_{\pm7.2}$ | (48) | | | | |
| | Gemma-12B | $3.1_{\pm3.5}$ | (500) | $96.0_{\pm6.0}$ | (485) | $48.7_{\pm20.6}$ | (19) | $92.3_{\pm18.8}$ | (11) | | | | |
| | Gemma-27B | $1.5_{\pm3.3}$ | (500) | $8.9_{\pm20.6}$ | (493) | $5.6_{\pm14.2}$ | (452) | $31.8_{\pm35.3}$ | (440) | | | | |
| | DeepSeek | $55.0_{\pm7.4}$ | (500) | $81.8_{\pm8.8}$ | (225) | $84.1_{\pm9.5}$ | (44) | $92.7_{\pm16.4}$ | (8) | | | | |
| **6 Rounds** | Humans | 1 | (281) | 6 | (279) | 21 | (261) | 53 | (205) | 73 | (97) | 85 | (26) |
| | LLaMA-3.2-3B | $34.6_{\pm13.6}$ | (500) | $20.1_{\pm9.9}$ | (327) | $22.6_{\pm14.7}$ | (266) | $21.3_{\pm16.7}$ | (215) | $17.4_{\pm14.2}$ | (181) | $19.2_{\pm12.3}$ | (158) |
| | LLaMA-3.1-8B | $38.4_{\pm4.8}$ | (500) | $38.9_{\pm3.8}$ | (308) | $60.8_{\pm4.0}$ | (189) | $55.3_{\pm7.0}$ | (74) | $58.8_{\pm8.2}$ | (33) | $43.0_{\pm15.2}$ | (14) |
| | LLaMA-Scout | $60.9_{\pm7.3}$ | (500) | $94.3_{\pm9.7}$ | (196) | $96.7_{\pm9.9}$ | (12) | $100.0_{\pm0.0}$ | (2) | - | (0) | - | (0) |
| | OLMo-13B | $23.5_{\pm13.2}$ | (500) | $24.1_{\pm17.9}$ | (382) | $48.2_{\pm21.5}$ | (299) | $41.9_{\pm23.0}$ | (174) | $21.0_{\pm13.6}$ | (124) | $26.5_{\pm21.3}$ | (108) |
| | OLMo-32B | $17.9_{\pm11.6}$ | (500) | $44.4_{\pm9.9}$ | (410) | $79.1_{\pm6.8}$ | (232) | $77.4_{\pm4.5}$ | (48) | $77.8_{\pm15.1}$ | (11) | $87.0_{\pm18.2}$ | (2) |
| | Gemma-12B | $3.4_{\pm3.1}$ | (500) | $88.8_{\pm23.5}$ | (483) | $51.0_{\pm23.8}$ | (55) | $33.2_{\pm46.8}$ | (40) | $0.0_{\pm0.0}$ | (6) | $20.0_{\pm44.7}$ | (6) |
| | Gemma-27B | $0.5_{\pm1.0}$ | (500) | $1.9_{\pm4.6}$ | (498) | $0.0_{\pm0.0}$ | (488) | $0.3_{\pm0.8}$ | (488) | $0.0_{\pm0.0}$ | (487) | $17.9_{\pm24.0}$ | (487) |
| | DeepSeek | $46.0_{\pm4.2}$ | (500) | $63.1_{\pm5.8}$ | (270) | $68.8_{\pm11.7}$ | (100) | $87.6_{\pm10.9}$ | (33) | $90.9_{\pm15.8}$ | (5) | $60.0_{\pm34.6}$ | (1) |
| **6 Rounds (H. Payoff)** | LLaMA-3.2-3B | $30.1_{\pm15.1}$ | (500) | $19.0_{\pm10.2}$ | (350) | $23.1_{\pm18.4}$ | (290) | $22.6_{\pm18.5}$ | (236) | $15.2_{\pm10.4}$ | (199) | $19.3_{\pm16.1}$ | (177) |
| | LLaMA-3.1-8B | $37.5_{\pm6.7}$ | (500) | $34.2_{\pm3.8}$ | (313) | $59.4_{\pm7.1}$ | (206) | $54.5_{\pm10.1}$ | (85) | $60.9_{\pm14.3}$ | (40) | $35.8_{\pm18.9}$ | (18) |
| | LLaMA-Scout | $65.1_{\pm6.2}$ | (500) | $88.7_{\pm10.0}$ | (175) | $89.7_{\pm20.6}$ | (21) | $34.8_{\pm7.3}$ | (4) | $100.0_{\pm0.0}$ | (3) | - | (0) |
| | OLMo-13B | $25.1_{\pm16.0}$ | (500) | $18.7_{\pm16.3}$ | (375) | $34.3_{\pm21.2}$ | (315) | $23.5_{\pm18.4}$ | (228) | $13.2_{\pm9.3}$ | (195) | $13.7_{\pm8.5}$ | (180) |
| | OLMo-32B | $17.0_{\pm10.1}$ | (500) | $44.2_{\pm6.7}$ | (415) | $75.1_{\pm8.9}$ | (234) | $73.3_{\pm5.3}$ | (58) | $79.9_{\pm14.7}$ | (15) | $89.9_{\pm13.3}$ | (4) |
| | Gemma-12B | $1.0_{\pm1.3}$ | (500) | $87.2_{\pm21.0}$ | (495) | $26.5_{\pm18.5}$ | (63) | $28.4_{\pm39.4}$ | (55) | $2.6_{\pm5.3}$ | (25) | $0.0_{\pm0.0}$ | (24) |
| | Gemma-27B | $0.6_{\pm2.0}$ | (500) | $1.5_{\pm3.6}$ | (497) | $0.6_{\pm1.9}$ | (490) | $0.2_{\pm0.4}$ | (487) | $0.0_{\pm0.0}$ | (487) | $4.6_{\pm10.0}$ | (487) |
| | DeepSeek | $44.0_{\pm5.4}$ | (500) | $65.0_{\pm5.9}$ | (280) | $71.2_{\pm12.0}$ | (99) | $76.7_{\pm17.8}$ | (29) | $89.2_{\pm14.3}$ | (8) | $60.4_{\pm42.7}$ | (2) |

Table 7: **CENTIPEDE GAME– Cumulative outcome probabilities.** $F_t = \sum_{j=1}^{t} e_j$, where $e_j$ is the proportion of games ending at round $j$. For the $(4, 1)$ payoff schedule we have $m_1 = 1$, $M_1 = 4$; when labeled *H.payoff* we use the high-stakes schedule $(16, 4)$ with $m_1 = 4$, $M_1 = 16$. Entries are *mean $\pm$ standard deviation* across verbalizations. Values typeset in green indicate an *increase* from rounds 1–5 to rounds 6–10; values in red indicate a *decrease*.

| $F_t$ (%) | | $F_1$ | | $F_2$ | | $F_3$ | | $F_4$ | | $F_5$ | | $F_6$ | |
|---|---|---|---|---|---|---|---|---|---|---|---|---|---|
| Rounds | | 1–5 | 6–10 | 1–5 | 6–10 | 1–5 | 6–10 | 1–5 | 6–10 | 1–5 | 6–10 | 1–5 | 6–10 |
| | Humans | 6.2 | 8.1 | 36.5 | 49.3 | 72.4 | 87.5 | 92.4 | 97.8 | | | | |
| | LLaMA-3.2-3B | $31.0_{\pm14.1}$ | $29.2_{\pm15.8}$ | $43.2_{\pm18.2}$ | $44.7_{\pm18.3}$ | $55.3_{\pm19.3}$ | $57.7_{\pm20.8}$ | $63.8_{\pm20.8}$ | $69.6_{\pm18.6}$ | | | | |
| | LLaMA-3.1-8B | $34.6_{\pm5.0}$ | $38.9_{\pm6.3}$ | $60.2_{\pm5.9}$ | $61.7_{\pm6.8}$ | $83.8_{\pm4.4}$ | $86.4_{\pm6.1}$ | $92.2_{\pm3.8}$ | $95.2_{\pm2.2}$ | | | | |
| 4 Rounds | LLaMA-4-Scout | $43.1_{\pm4.2}$ | $80.1_{\pm11.2}$ | $98.6_{\pm1.8}$ | $99.6_{\pm0.7}$ | $100.0_{\pm0.0}$ | $100.0_{\pm0.0}$ | $100.0_{\pm0.0}$ | $100.0_{\pm0.0}$ | | | | |
| | OLMo-13B | $26.2_{\pm12.3}$ | $25.4_{\pm13.2}$ | $45.2_{\pm20.8}$ | $47.2_{\pm22.0}$ | $72.6_{\pm15.8}$ | $75.5_{\pm17.0}$ | $85.2_{\pm13.1}$ | $87.0_{\pm13.5}$ | | | | |
| | OLMo-32B | $17.2_{\pm7.3}$ | $25.0_{\pm15.0}$ | $51.3_{\pm8.7}$ | $68.7_{\pm11.3}$ | $86.9_{\pm7.2}$ | $96.2_{\pm3.2}$ | $97.0_{\pm2.7}$ | $99.5_{\pm0.5}$ | | | | |
| | Gemma-12B | $0.6_{\pm0.8}$ | $13.4_{\pm7.2}$ | $98.4_{\pm3.7}$ | $98.7_{\pm3.0}$ | $99.6_{\pm0.9}$ | $99.8_{\pm0.5}$ | $100.0_{\pm0.1}$ | $100.0_{\pm0.0}$ | | | | |
| | Gemma-27B | $1.5_{\pm3.3}$ | $5.8_{\pm13.4}$ | $11.2_{\pm23.1}$ | $20.1_{\pm39.5}$ | $11.8_{\pm24.4}$ | $23.0_{\pm39.3}$ | $55.6_{\pm36.7}$ | $64.5_{\pm42.1}$ | | | | |
| | DeepSeek | $41.5_{\pm4.5}$ | $62.8_{\pm5.7}$ | $82.0_{\pm6.7}$ | $97.9_{\pm1.4}$ | $96.7_{\pm2.4}$ | $100.0_{\pm0.0}$ | $99.8_{\pm0.3}$ | $100.0_{\pm0.0}$ | | | | |
| | LLaMA-3.2-3B | $26.9_{\pm10.2}$ | $22.3_{\pm12.1}$ | $39.6_{\pm15.9}$ | $41.1_{\pm17.0}$ | $53.8_{\pm16.3}$ | $55.1_{\pm19.7}$ | $63.2_{\pm17.4}$ | $66.6_{\pm20.3}$ | | | | |
| | LLaMA-3.1-8B | $34.4_{\pm5.4}$ | $39.2_{\pm5.4}$ | $58.6_{\pm5.6}$ | $61.0_{\pm6.2}$ | $81.6_{\pm5.0}$ | $85.0_{\pm5.5}$ | $90.3_{\pm3.5}$ | $94.9_{\pm2.5}$ | | | | |
| 4 Rounds (H. payoff) | LLaMA-4-Scout | $43.8_{\pm4.6}$ | $90.7_{\pm11.2}$ | $98.1_{\pm2.5}$ | $100.0_{\pm0.1}$ | $99.8_{\pm0.6}$ | $100.0_{\pm0.0}$ | $100.0_{\pm0.0}$ | $100.0_{\pm0.0}$ | | | | |
| | OLMo-13B | $26.6_{\pm13.6}$ | $26.4_{\pm14.8}$ | $39.9_{\pm20.1}$ | $43.4_{\pm23.9}$ | $64.1_{\pm22.4}$ | $65.5_{\pm26.9}$ | $74.8_{\pm21.1}$ | $74.9_{\pm26.3}$ | | | | |
| | OLMo-32B | $19.0_{\pm7.4}$ | $26.0_{\pm14.7}$ | $52.3_{\pm9.3}$ | $70.0_{\pm12.7}$ | $85.4_{\pm4.2}$ | $95.2_{\pm4.1}$ | $96.4_{\pm2.3}$ | $99.5_{\pm0.6}$ | | | | |
| | Gemma-12B | $0.4_{\pm0.9}$ | $5.7_{\pm6.4}$ | $96.0_{\pm6.0}$ | $96.4_{\pm5.3}$ | $96.6_{\pm5.6}$ | $98.9_{\pm2.1}$ | $99.8_{\pm0.6}$ | $100.0_{\pm0.1}$ | | | | |
| | Gemma-27B | $0.6_{\pm1.4}$ | $2.3_{\pm5.5}$ | $5.5_{\pm14.4}$ | $13.7_{\pm29.9}$ | $8.6_{\pm21.4}$ | $15.4_{\pm32.7}$ | $32.3_{\pm36.2}$ | $39.5_{\pm44.1}$ | | | | |
| | DeepSeek | $43.8_{\pm7.7}$ | $66.3_{\pm7.7}$ | $83.9_{\pm9.0}$ | $98.6_{\pm1.5}$ | $96.8_{\pm2.7}$ | $100.0_{\pm0.0}$ | $99.8_{\pm0.3}$ | $100.0_{\pm0.0}$ | | | | |
| | Humans | 0.0 | 1.5 | 5.5 | 8.9 | 22.7 | 31.7 | 55.8 | 75.8 | 88.9 | 92.7 | 97.9 | 99.3 |
| | LLaMA-3.2-3B | $32.0_{\pm12.1}$ | $37.1_{\pm15.7}$ | $42.5_{\pm15.3}$ | $51.0_{\pm17.0}$ | $53.7_{\pm17.4}$ | $60.3_{\pm19.5}$ | $61.1_{\pm19.1}$ | $66.5_{\pm20.2}$ | $66.4_{\pm18.5}$ | $70.3_{\pm20.3}$ | $70.8_{\pm18.2}$ | $74.4_{\pm19.7}$ |
| | LLaMA-3.1-8B | $32.9_{\pm5.7}$ | $44.0_{\pm4.7}$ | $60.8_{\pm5.3}$ | $63.7_{\pm5.3}$ | $83.4_{\pm3.4}$ | $86.8_{\pm3.5}$ | $91.3_{\pm2.7}$ | $95.4_{\pm1.7}$ | $96.1_{\pm1.7}$ | $98.2_{\pm0.9}$ | $97.4_{\pm1.6}$ | $99.2_{\pm0.4}$ |
| 6 Rounds | LLaMA-4-Scout | $42.1_{\pm5.3}$ | $79.6_{\pm10.6}$ | $97.0_{\pm6.3}$ | $98.2_{\pm3.4}$ | $99.2_{\pm2.4}$ | $100.0_{\pm0.0}$ | $100.0_{\pm0.0}$ | $100.0_{\pm0.0}$ | $100.0_{\pm0.0}$ | $100.0_{\pm0.0}$ | $100.0_{\pm0.0}$ | $100.0_{\pm0.0}$ |
| | OLMo-13B | $22.4_{\pm12.3}$ | $24.6_{\pm14.5}$ | $37.3_{\pm20.4}$ | $43.1_{\pm23.8}$ | $63.2_{\pm23.5}$ | $67.3_{\pm25.8}$ | $74.6_{\pm24.2}$ | $75.6_{\pm26.0}$ | $79.2_{\pm21.5}$ | $77.6_{\pm24.6}$ | $82.3_{\pm20.2}$ | $79.4_{\pm24.4}$ |
| | OLMo-32B | $14.4_{\pm7.5}$ | $21.4_{\pm16.1}$ | $43.6_{\pm10.1}$ | $63.7_{\pm16.0}$ | $83.6_{\pm6.3}$ | $97.3_{\pm1.6}$ | $95.9_{\pm1.9}$ | $99.7_{\pm0.3}$ | $99.1_{\pm0.8}$ | $99.8_{\pm0.2}$ | $99.8_{\pm0.2}$ | $100.0_{\pm0.0}$ |
| | Gemma-12B | $0.1_{\pm0.2}$ | $6.7_{\pm6.2}$ | $88.8_{\pm23.5}$ | $89.2_{\pm22.5}$ | $91.0_{\pm21.6}$ | $93.1_{\pm19.4}$ | $98.4_{\pm3.0}$ | $99.3_{\pm1.4}$ | $98.4_{\pm3.0}$ | $99.3_{\pm1.4}$ | $98.4_{\pm3.0}$ | $99.4_{\pm1.4}$ |
| | Gemma-27B | $0.2_{\pm0.4}$ | $0.8_{\pm1.6}$ | $1.2_{\pm2.9}$ | $3.4_{\pm7.8}$ | $1.2_{\pm2.9}$ | $3.4_{\pm7.8}$ | $1.8_{\pm4.2}$ | $3.4_{\pm7.8}$ | $1.8_{\pm4.2}$ | $3.4_{\pm7.8}$ | $18.3_{\pm21.8}$ | $21.7_{\pm26.2}$ |
| | DeepSeek | $35.2_{\pm4.1}$ | $56.8_{\pm5.2}$ | $64.9_{\pm6.9}$ | $94.9_{\pm2.0}$ | $87.0_{\pm6.6}$ | $99.9_{\pm0.2}$ | $97.9_{\pm2.9}$ | $100.0_{\pm0.0}$ | $99.6_{\pm0.8}$ | $100.0_{\pm0.0}$ | $99.8_{\pm0.5}$ | $100.0_{\pm0.0}$ |
| | LLaMA-3.2-3B | $29.1_{\pm13.5}$ | $31.1_{\pm17.6}$ | $38.9_{\pm17.0}$ | $45.3_{\pm19.5}$ | $50.2_{\pm20.4}$ | $55.3_{\pm23.1}$ | $58.1_{\pm22.5}$ | $62.1_{\pm23.8}$ | $63.2_{\pm21.8}$ | $65.9_{\pm23.3}$ | $67.4_{\pm21.9}$ | $70.3_{\pm22.9}$ |
| | LLaMA-3.1-8B | $33.2_{\pm6.7}$ | $41.8_{\pm7.2}$ | $56.5_{\pm6.7}$ | $61.1_{\pm5.1}$ | $80.4_{\pm5.5}$ | $85.6_{\pm5.2}$ | $89.6_{\pm5.2}$ | $94.2_{\pm3.4}$ | $95.0_{\pm3.2}$ | $98.0_{\pm2.0}$ | $96.6_{\pm2.4}$ | $99.0_{\pm1.2}$ |
| 6 Rounds (H. payoff) | LLaMA-4-Scout | $41.9_{\pm5.2}$ | $88.3_{\pm8.5}$ | $94.2_{\pm7.4}$ | $97.3_{\pm3.7}$ | $98.3_{\pm3.7}$ | $100.0_{\pm0.0}$ | $100.0_{\pm0.0}$ | $100.0_{\pm0.0}$ | $100.0_{\pm0.0}$ | $100.0_{\pm0.0}$ | $100.0_{\pm0.0}$ | $100.0_{\pm0.0}$ |
| | OLMo-13B | $24.1_{\pm15.5}$ | $26.0_{\pm16.6}$ | $35.3_{\pm22.0}$ | $38.6_{\pm24.8}$ | $53.2_{\pm26.1}$ | $55.8_{\pm29.4}$ | $59.9_{\pm28.1}$ | $62.2_{\pm31.9}$ | $63.6_{\pm28.7}$ | $64.2_{\pm31.8}$ | $68.6_{\pm26.8}$ | $66.2_{\pm31.1}$ |
| | OLMo-2-32B | $14.3_{\pm7.9}$ | $19.7_{\pm13.3}$ | $42.2_{\pm8.1}$ | $64.2_{\pm12.4}$ | $80.6_{\pm7.0}$ | $96.4_{\pm2.7}$ | $94.3_{\pm2.5}$ | $99.6_{\pm0.3}$ | $98.6_{\pm1.3}$ | $100.0_{\pm0.1}$ | $99.8_{\pm0.3}$ | $100.0_{\pm0.0}$ |
| | Gemma-12B | $0.4_{\pm0.9}$ | $1.5_{\pm1.9}$ | $87.2_{\pm20.9}$ | $87.4_{\pm20.4}$ | $87.7_{\pm20.9}$ | $90.2_{\pm19.0}$ | $94.3_{\pm9.1}$ | $95.7_{\pm8.0}$ | $94.4_{\pm9.0}$ | $95.9_{\pm7.7}$ | $94.4_{\pm9.0}$ | $95.9_{\pm7.7}$ |
| | Gemma-27B | $0.4_{\pm1.3}$ | $0.9_{\pm2.8}$ | $1.5_{\pm4.0}$ | $2.6_{\pm6.7}$ | $2.0_{\pm5.5}$ | $3.1_{\pm8.4}$ | $2.2_{\pm6.1}$ | $3.1_{\pm8.4}$ | $2.2_{\pm6.1}$ | $3.1_{\pm8.4}$ | $5.9_{\pm13.3}$ | $7.1_{\pm16.2}$ |
| | DeepSeek | $31.0_{\pm5.2}$ | $57.0_{\pm6.4}$ | $64.0_{\pm7.7}$ | $96.4_{\pm2.4}$ | $88.6_{\pm5.3}$ | $100.0_{\pm0.1}$ | $96.8_{\pm3.3}$ | $100.0_{\pm0.0}$ | $99.3_{\pm1.2}$ | $100.0_{\pm0.0}$ | $99.8_{\pm0.4}$ | $100.0_{\pm0.0}$ |

Table 8: **CENTIPEDE GAME– Expected payoffs of players.** Asterisks mark one-sided sign-flip permutation tests *vs.* humans.

| $\overline{\pi}_i$ | 4 Rounds | | 4 Rounds (H. P.) | | 6 Rounds | | 6 Rounds (H. P.) | |
|---|---|---|---|---|---|---|---|---|
| | $\overline{\pi}_1$ | $\overline{\pi}_2$ | $\overline{\pi}_1$ | $\overline{\pi}_2$ | $\overline{\pi}_1$ | $\overline{\pi}_2$ | $\overline{\pi}_1$ | $\overline{\pi}_2$ |
| Humans | 11.38 | 10.08 | 37.68 | 36.12 | 28.72 | 28.67 | - | - |
| LLaMA-3.2-3B | 25.61* | 10.51 | 107.36* | 44.14 | 78.16* | 27.65 | 350.88 | 120.70 |
| LLaMA-3.1-8B | 10.55 | 7.03 | 44.67 | 29.23 | 13.51 | 8.97 | 61.97 | 38.81 |
| LLaMA-Scout | 3.35 | 3.65 | 13.89 | 13.13 | 3.52 | 3.75 | 16.58 | 13.68 |
| OLMo-13B | 15.76 | 9.08 | 87.79* | 38.63 | 58.00 | 21.65 | 363.68 | 119.25 |
| OLMo-32B | 8.31 | 7.02 | 33.01 | 29.06 | 9.33 | 7.70 | 38.70 | 33.59 |
| Gemma-12B | 2.33 | 7.53 | 9.94 | 32.96 | 5.71 | 9.96 | 59.97 | 48.15 |
| Gemma-27B | 29.66* | 21.10* | 174.20* | 74.67* | 210.46* | 73.76* | 962.93 | 259.75 |
| DeepSeek | 4.36 | 4.40 | 16.95 | 16.96 | 6.00 | 5.82 | 24.84 | 23.23 |
| NASH EQ. | 4.00 | 1.00 | 16.00 | 4.00 | 4.00 | 1.00 | 16.00 | 4.00 |

$p$-value $< 0.05$ ($H_1: \overline{\pi}_i^{(\text{LLM})} > \overline{\pi}_i^{(\text{human})}$)

### E.3 TRAVELER'S DILEMMA: ADDITIONAL RESULTS

Table 9: **TRAVELER'S DILEMMA: Average claims (symmetric $r$).** For each bonus/penalty $r \in \{5, 10, 20, 25, 50, 80\}$, columns "1" and "8–10" report the average claim in round 1 and in the last three rounds of the session, respectively. Entries are *mean $\pm$ standard deviation* across verbalizations. Values typeset in green indicate an *increase* from round 1 to rounds 8–10; values in red indicate a *decrease*. Human results are shown for reference. NASH EQ. and PARETO EFF. provide theoretical benchmarks.

| | $r = 5$ | | $r = 10$ | | $r = 20$ | | $r = 25$ | | $r = 50$ | | $r = 80$ | |
|---|---|---|---|---|---|---|---|---|---|---|---|---|
| | 1 | 8-10 | 1 | 8-10 | 1 | 8-10 | 1 | 8-10 | 1 | 8-10 | 1 | 8-10 |
| Humans | 180 | 195 | 177 | 186 | 131 | 119 | 150 | 138 | 155 | 85 | 120 | 81 |
| LLaMA-3.2-3B | $146_{\pm33}$ | $169_{\pm28}$ | $148_{\pm29}$ | $168_{\pm26}$ | $144_{\pm25}$ | $166_{\pm28}$ | $147_{\pm28}$ | $165_{\pm29}$ | $148_{\pm24}$ | $163_{\pm29}$ | $148_{\pm26}$ | $165_{\pm29}$ |
| LLaMA-3.1-8B | $168_{\pm32}$ | $179_{\pm27}$ | $168_{\pm30}$ | $181_{\pm28}$ | $164_{\pm30}$ | $174_{\pm31}$ | $169_{\pm30}$ | $177_{\pm27}$ | $172_{\pm31}$ | $178_{\pm30}$ | $184_{\pm25}$ | $174_{\pm33}$ |
| LLaMA-Scout | $116_{\pm32}$ | $198_{\pm9}$ | $148_{\pm7}$ | $176_{\pm19}$ | $130_{\pm14}$ | $187_{\pm23}$ | $88_{\pm20}$ | $199_{\pm11}$ | $149_{\pm5}$ | $200_{\pm0}$ | $80_{\pm0}$ | $165_{\pm54}$ |
| OLMo-13B | $108_{\pm31}$ | $170_{\pm29}$ | $113_{\pm29}$ | $174_{\pm28}$ | $109_{\pm25}$ | $168_{\pm29}$ | $111_{\pm23}$ | $170_{\pm28}$ | $119_{\pm26}$ | $171_{\pm23}$ | $130_{\pm32}$ | $168_{\pm27}$ |
| OLMo-32B | $140_{\pm44}$ | $167_{\pm36}$ | $142_{\pm44}$ | $164_{\pm35}$ | $144_{\pm40}$ | $169_{\pm30}$ | $144_{\pm45}$ | $174_{\pm32}$ | $136_{\pm44}$ | $163_{\pm34}$ | $147_{\pm36}$ | $159_{\pm31}$ |
| Gemma-12B | $96_{\pm12}$ | $97_{\pm12}$ | $101_{\pm5}$ | $108_{\pm17}$ | $96_{\pm20}$ | $99_{\pm23}$ | $103_{\pm11}$ | $109_{\pm17}$ | $117_{\pm15}$ | $122_{\pm20}$ | $101_{\pm11}$ | $104_{\pm16}$ |
| Gemma-27B | $80_{\pm2}$ | $80_{\pm2}$ | $149_{\pm54}$ | $144_{\pm53}$ | $100_{\pm1}$ | $100_{\pm3}$ | $100_{\pm1}$ | $101_{\pm3}$ | $120_{\pm14}$ | $137_{\pm32}$ | $121_{\pm8}$ | $121_{\pm8}$ |
| DeepSeek | $154_{\pm58}$ | $117_{\pm55}$ | $150_{\pm59}$ | $108_{\pm51}$ | $129_{\pm57}$ | $101_{\pm43}$ | $154_{\pm56}$ | $126_{\pm56}$ | $110_{\pm43}$ | $86_{\pm24}$ | $130_{\pm54}$ | $89_{\pm28}$ |
| NASH EQ. | 80 | 80 | 80 | 80 | 80 | 80 | 80 | 80 | 80 | 80 | 80 | 80 |
| PARETO EFF. | 200 | 200 | 200 | 200 | 200 | 200 | 200 | 200 | 200 | 200 | 200 | 200 |

Table 10: **TRAVELER'S DILEMMA– Average claims (asymmetric $r$).** Each column $r_1$-$r_2$ reports player 1's average claim. Symmetric columns come from baseline runs with $r=10$ and $r=80$, pooling both players. Bold marks the larger mean within each pair.

|  | Asymmetric | | Symmetric | |
|---|---|---|---|---|
|  | 10–80 | 80–10 | 10–10 | 80–80 |
| Humans | **149.1** | 112.3 | **174.4** | 115.6 |
| LLaMA-3.2-3B | **150.3***  | 137.9 | **147.8** | 147.6 |
| LLaMA-3.1-8B | 169.4 | **181.2** | 168.2 | **183.9** |
| LLaMA-Scout | 83.4 | **84.1** | **148.4** | 80.0 |
| OLMo-13B | 122.2† | **131.8** | 112.5 | **129.8** |
| OLMo-32B | **154.2**† | 148.1 | 141.8 | **146.6** |
| Gemma-12B | 99.5 | **104.0** | 100.6 | **101.0** |
| Gemma-27B | **183.8***† | 131.6 | **149.2** | 121.2 |
| DeepSeek | **160.1** | 144.2 | **150.2** | 130.1 |
| NASH EQ. | 80.0 | 80.0 | 80.0 | 80.0 |
| PARETO EFF. | 200.0 | 200.0 | 200.0 | 200.0 |

\* $p$-value $< 0.05$ ($H_1$: 10-80 > 80-10)

† $p$-value $< 0.05$ ($H_1$: 10-80 > 10-10)

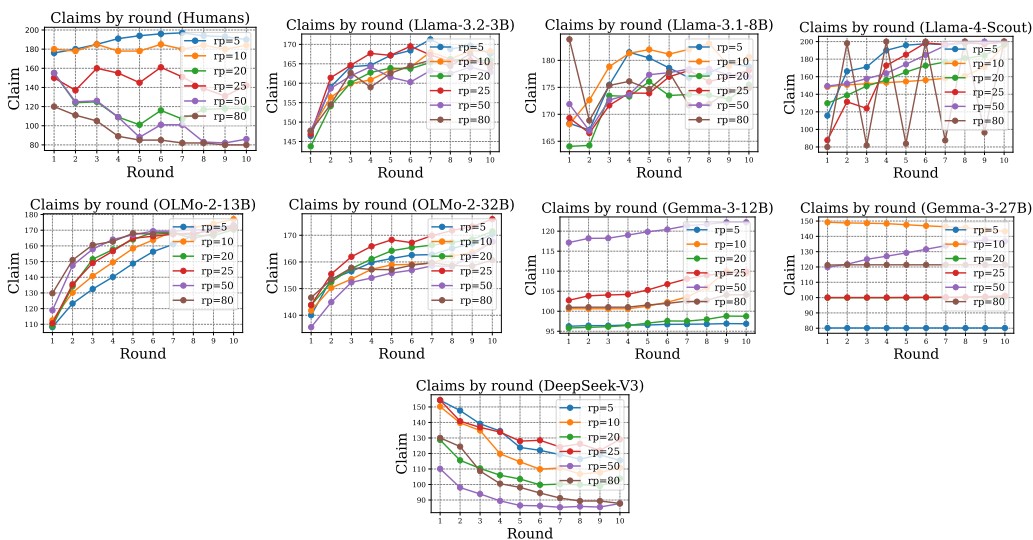

Figure 7: **TRAVELER'S DILEMMA– Claims by round (extended).** Average claim across rounds for human players (top left, (Capra et al., 1999)) and for each LLM, under different bonus/penalty values $r_p \in \{5, 10, 20, 25, 50, 80\}$. Curves show how claims evolve within a 10-round session. Similar to humans, many models display systematic trends (increasing or decreasing) as the game progresses, while others remain relatively flat. This figure extends Fig. 7 by showing results for all models.

## F PROMPT TEMPLATES

### F.1 ROCK, PAPER, SCISSORS

```
1 {
2     "system": (
3         "You are a player in a game that lasts for {round_num} repeated rounds. "
4         "You will interact with a different player at each round. "
5         "At each round, you must emit exactly one symbol from the set: {rock}, {paper},
          ↪ {scissors}. "
6         "Rules: {rock} defeats {scissors}, {scissors} defeats {paper}, and {paper}
          ↪ defeats {rock}. "
7         "If your symbol defeats the other player's, you receive {a:.1f} points, the other
          ↪ player receives 0. "
```

```
 8          "If you are defeated, you receive 0 and the other player receives {a:.1f}. "
 9          "If symbols are identical, both players receive 1 point. "
10          "Output constraint: reply EXACTLY 'I choose <symbol>', with <symbol> one of
     ↪      {rock}, {paper}, {scissors}. "
11          "Your objective is to maximize your payoff at each round and overall. "
12      )
13 }
14 {
15      "user": (
16          "Round {round_id}/{n_rounds}. Which option do you choose?"
17      )
18 }
19 {
20      "assistant": "I choose {option}"
21 }
22 {
23      "user": (
24          "In round {round_id}, you chose {self_option}, the other player chose
     ↪      {other_option}. "
25          "Thus, you receive {self_gain} and the other player receives {other_gain}."
26      )
27 }
```

## F.2   CENTIPEDE GAME

```
28 {
29      "system": (
30          "Scenario: Two players interact for up to {N_ROUNDS} rounds with two numbers {X}
     ↪       and {Y}. "
31          "On each round, the acting player must output exactly one token: {TAKE} or {PASS}.
     ↪      "
32          "· If {TAKE}: the acting player receives {X} {UNIT}, the other player receives
     ↪      {Y} {UNIT}; the interaction ends. "
33          "· If {PASS}: {INCREASE_PROMPT}, and control passes to the other player. "
34          "· If no one takes before the final round, the final mover must TAKE: they
     ↪       receive {X} {UNIT}, the other player receives {Y} {UNIT}. "
35          "After the final round, the interaction ends. A new interaction begins with {X},
     ↪      {Y} reset. "
36          "There are {TOTAL_GAMES} interactions in total. "
37          "Objective: maximize your payoff within each interaction and across all
     ↪       interactions. "
38          "You are player {player}. This is the beginning of the first interaction."
39      )
40 }
41 {
42      "user": (
43          "Round {ROUND_ID}: the numbers are {X} and {Y}. "
44          "Decide: {TAKE} or {PASS}? Reply EXACTLY 'I choose <option>'."
45      )
46 }
47 {
48      "assistant": "I choose {option}"
49 }
50 {
51      "user": (
52          "The other player chose {other_option} in round {PREV_ROUND_ID}. "
53          "Outcome: you receive {self_gain}, the other player receives {other_gain}. "
54          "Prepare for the next interaction ({INT_N}/{TOTAL_GAMES}), starting with round
     ↪      {ROUND_ID} and numbers {X}, {Y}."
55      )
56 }
```

## F.3   TRAVELER'S DILEMMA

```
57 {
58      "system": (
```

```
59        "You are a player in a game of {n_rounds} rounds. "
60        "In each round, you and another player simultaneously select an integer k in [80,
   ↪  {max_value}]. "
61        "Payoff rules: "
62        "1. If both select the same k, each receives k. "
63        "2. If your k < opponent's k: you receive k + {bp_value_1}, opponent receives k -
   ↪  {bp_value_2}. "
64        "3. If your k > opponent's k: you receive opponent's k - {bp_value_1}, opponent
   ↪   receives opponent's k + {bp_value_2}. "
65        "Your role: maximize your payoff each round and across the game. "
66        "Output constraint: reply EXACTLY 'I choose <k>', where <k> is an integer in [80,
   ↪  {max_value}]."
67    )
68 }
69 {
70    "user": (
71        "Round {round_id}/{n_rounds}. What integer value do you select?"
72    )
73 }
74 {
75    "assistant": "I choose {option}"
76 }
77 {
78    "user": (
79        "In round {round_id}, you chose {self_option}, the other player chose
   ↪  {other_option}. "
80        "Thus, you receive {self_gain}, the other player receives {other_gain}."
81    )
82 }
```

