# OpenReview forum: "Rational Irrationality: Evaluating LLMs In Games With Strategic Behavior Discrepancies"
_ICLR.cc/2026/Conference — Submitted to ICLR 2026_

### Official Review · Reviewer_UBaE · 2025-10-23

**Soundness:** 2
**Presentation:** 3
**Contribution:** 2
**Rating:** 2
**Confidence:** 4

**Summary:**

This paper evaluates how LLMs make strategic decisions by comparing their behavior to both game-theoretic predictions and human experimental data. Using three classic games, i.e., Rock-Paper-Scissors, Centipede, and Traveler’s Dilemma, the authors find that LLMs generally behave more rationally than humans, aligning more closely with theoretical equilibria. However, they show limited sensitivity to payoff changes and exhibit prompt-dependent variability, indicating shallow or unstable strategic reasoning. Overall, the study reveals that LLMs tend to favor safe, equilibrium-like strategies over the cooperative or risk-taking behaviors typically observed in humans.

**Strengths:**

1. They have open-source code for better reproducibility
2. Evaluate prompt sensitivity
3. Collect documented human behaviors for reference

**Weaknesses:**

1. Only open-source models are evaluated. Could you please also evaluate some common proprietary models like GPT, Gemini, and Claude for an easier comparison with previous work?
2. Game selection. The current three games are all two-person zero-sum or quasi-zero-sum games. The conclusion may be limited. For example, the conclusion may not generalize to cooperative games. The limitation should be reflected either in a limitation section, the introduction, or in the title.
3. The key conclusion in RPS is that LLMs do not align with NE, which is different from some previous findings where LLM choices are very skewed toward rock [1]. Could you please explain?
4. In my understanding, the prompt variants differ only in the symbols. There are not any rephrases. I think the change is very minimal and may not reflect the actual prompt sensitivity.
5. Studying whether LLMs prefer optimal strategies or average human choices has been studied in this area. Some papers are not discussed sufficiently, especially for those with different conclusions. For example, Brookins and DeBacker, 2023 concluded that “the LLM replicates human tendencies towards fairness and cooperation. It does not choose the optimal strategy in most cases.” Additionally, Huang et al, (2025 concluded that “its predictions are more aligned with human behavior than the game’s NE,” in section 4.3. I think more discussions with these conclusions should be put into the paper.

[1] Language Agents with Reinforcement Learning for Strategic Play in the Werewolf Game. ICML 2024.

**Questions:**

1. Line 105: “This results in models performing better than the NE strategy, but worse than humans.”
2. Line 253: “the number of participants to 100.” Do you randomly pair the 100 LLMs into 50 pairs for each round?
3. DeepSeek-V3 performs differently in CG and TD, one against NE and the other toward. How to understand such a discrepancy?
4. Do you examine whether LLMs know the optimal strategy for each game? You can simply ask LLMs to analyze and see whether their answers are correct. Also, you can vary the game settings and rephrase the sentences to avoid game leakage.
5. I really want to see more discussions about the broader impact of LLMs if they align more with GT-predicted optimal strategies and are used in some financial decision scenarios.
6. The human references are from 1999, 1992, and 2014. They are 20~30 years ago. How much do you think human behaviors nowadays have changed?
7. How will models behave if you apply CoT? A deeper analysis of LLM behaviors using their CoT is needed for this paper.

Minor suggestions and typos:
1. Fig 4: figures are too small.
2. Table 1: can add a row of “random.” Will it be all 33.3?
3. Table 1: the reference to Figure 2 should be Figure 1?

---

### Official Review · Reviewer_g9qD · 2025-10-28

**Soundness:** 2
**Presentation:** 3
**Contribution:** 1
**Rating:** 2
**Confidence:** 4

**Summary:**

The paper examines large language models (LLMs) in three classic game-theoretic environments—Rock-Paper-Scissors, Centipede Game, and Traveler’s Dilemma—to compare their strategic behavior against both game-theoretic (GT) rationality and empirical human data. The authors conclude that LLMs generally behave more rationally according to GT predictions than humans, are relatively insensitive to payoff hyperparameters, and sometimes exhibit partial human-like traits.

**Strengths:**

1. The experimental design is **systematic and reproducible**, with clear descriptions of protocols, baselines, and data sources.
2. The paper includes **comprehensive empirical comparisons** across several models and games, supported by detailed statistical results.

**Weaknesses:**

1. The **technical novelty is minimal.** The work largely reproduces existing studies on LLM behavior in classic games without introducing new theoretical, methodological, or empirical contributions.
2. The **research motivation is weak**, providing little justification for the significance or necessity of re-evaluating LLMs on these already well-studied game scenarios.
3. The **analysis remains descriptive**, lacking deeper interpretation of why certain models behave as they do or how prompt sensitivity influences outcomes.
4. The **paper is overly lengthy and unbalanced**, spending excessive space on textbook-level explanations of game theory while offering limited new insights.

**Questions:**

1. Rock–Paper–Scissors and Traveler’s Dilemma have been widely explored in prior LLM and behavioral-game-theory studies, whereas the Centipede Game is less examined. Could the authors clarify the rationale for choosing exactly these three games, and what new insight their combination provides beyond existing findings?
2. The paper reports notable variance across verbalizations but provides limited interpretation. Could the authors elaborate on what this variability indicates about model reasoning consistency, symbolic sensitivity, or policy stability? Would additional ablations (e.g., varying label semantics or framing) help disentangle these effects?
3. While Appendix D.1 briefly mentions temperature settings, the paper does not examine how temperature, model size, or multi-round context length affect behavioral stability. Would a systematic sensitivity analysis help confirm the robustness of the reported findings?

---

### Official Review · Reviewer_zhgV · 2025-10-31

**Soundness:** 2
**Presentation:** 2
**Contribution:** 1
**Rating:** 2
**Confidence:** 4

**Summary:**

In this paper, the authors examine the alignment between LLMs and humans in strategic behavior through a game-theoretic perspective. Specifically, they evaluate LLM performance in three games where empirical findings diverge from theoretical predictions. Their analysis indicates that LLMs tend to imitate rational behavior, blending sometimes with human-like strategies. Moreover, LLMs appear mostly insensitive to numerical payoff-related hyperparameters in prompts, contrasting with typical human behavior.

**Strengths:**

1. In this paper, the authors examine the alignment between LLMs and humans in strategic behavior through a game-theoretic perspective.
2. The authors conduct several experiment on this topic, including LLMs like Llama-3.2-3B, Llama-3.1-8B, Llama-4-Scout, OLMo-2-13B, OLMo-2-32B, Gemma-3-12B, Gemma-3-27B, and DeepSeek-V3.

**Weaknesses:**

1. For me, using game theory to evaluate large models is an ancient topic, and this article should have appeared two years ago instead of now. Therefore, in my view, this paper lack of novelty in some extend. The authors could this paper with some old paper, like Economics Arena for Large Language Models
2. In my view, I don't see the necessity of the alignment of LLMs with empirical human reasoning, why we need to do that, since humans are also irrational, why should we make LLMs irrational? Shouldn't we look for a way to make LLMs more rational?
3. In my view, using game theory alone to evaluate LLM is trivial and simple. How to make LLMs win in the game is important and difficult.

**Questions:**

Na

---

### Official Review · Reviewer_A1QA · 2025-11-05

**Soundness:** 2
**Presentation:** 2
**Contribution:** 3
**Rating:** 4
**Confidence:** 3

**Summary:**

Large language models (LLMs) are increasingly deployed in complex decisionmaking environments. Recently researchers are focusing on purely game theory theoretical expectations, but this work analysis in other way that evaluates the strategic reasoning of Large Language Models (LLMs) by assessing their alignment with human strategic thinking. The authors test LLMs in three specific games - the ROCK,PAPER,SCISSORS(RPS),the CENTIPEDE GAME (CG), and the TRAVELER'S DILEMMA(TD) — which are all characterized by significant discrepancies between game-theoretic predictions and actual human behavior. Based on the study, the experimental results indicate that LLM behavior aligns more closely with game-theoretic rationality than with human strategies. Furthermore, the study found that LLMs exhibit limited adaptability to changes in payoff-related game hyperparameters, a trait that contrasts with typical human behavior.

**Strengths:**

S1. The perspective from which this job discusses issues is very novel.
S2. The paper is built on a thorough theoretical grounding, clearly contrasting classical game-theoretic predictions with established findings from human-subject experiments.
S3. The experimental design is innovative, notably using multiple "verbalizations" to control for and measure the impact of prompt phrasing on strategic choices.

**Weaknesses:**

W1. The motivation and conclusions lack clear directive significance, offering little guiding value for future research.
W2. The models used in the experiment have relatively lagging capabilities and do not represent the latest or most powerful available models.
W3. The model selection is limited to open-source models and omits comparisons with leading closed-source counterparts.

**Questions:**

Q1. In the Traveler's Dilemma (TD), why did the LLMs fail to follow any consistent trend under asymmetric conditions, and how can this phenomenon be explained?
Q2. Have you attempted to use any reasoning models to analyze how the models are actually making these decisions?
Q3. What are the internal factors that cause different models to make different decisions? Furthermore, when different parameter-sized versions of the same model produce different results, what is the underlying cause for this discrepancy?

---

### Meta-Review · Area_Chair_QFZ1 · 2026-01-07

**Summary:**

Little support for acceptance in the original reviews (scores 4, 2, 2, 2). Authors have not provided responses. The paper is rejected.

**Reviewer Scores:**

No changes.

---

### Decision · Program_Chairs · 2026-01-26

Reject